



# Methane fluxes in the high northern latitudes for 2005 – 2013 estimated using a Bayesian atmospheric inversion

Rona L. Thompson[1], Motoki Sasakawa[2], Toshinobu Machida[2], Tuula Aalto[3], Doug Worthy[4],

Jost V. Lavric[5,6], Cathrine Lund Myhre[1] and Andreas Stohl[1]

[1]NILU - Norwegian Institute for Air Research, Kjeller, Norway

[2]National Institute for Environmental Studies, Tsukuba, Japan

[3]Finnish Meteorological Institute (FMI), Helsinki, Finland

[4]Environment Canada, Toronto, Canada

[5]Max Planck Institute for Biogeochemistry, Jena, Germany

[6]Integrated Carbon Observation System (ICOS), ERIC Head Office, Helsinki, Finland

*Corresponding author*: R. L. Thompson (rlt@nilu.no)

**Abstract**

We present methane ($CH_4$) flux estimates for 2005 to 2013 from a Bayesian inversion focusing on the high northern latitudes (north of 50°N). Our inversion is based on atmospheric transport modelled by the Lagrangian particle dispersion model, FLEXPART,

and $CH_4$ observations from 17 in-situ and 5 discrete flask-sampling sites distributed over northern North America and Eurasia. $CH_4$ fluxes are determined at monthly temporal resolution and on a variable grid with maximum resolution of 1° × 1°. Our inversion finds a $CH_4$ source from the high northern latitudes of 82 to 84 Tg y$^{-1}$, constituting ~15% of the global total, compared to 64 to 68 Tg y$^{-1}$ (~12%) in the prior estimates. For northern North

America, we estimate a mean source of 16.6 to 17.9 Tg y$^{-1}$, which is dominated by fluxes in the Hudson Bay Lowlands (HBL) and western Canada, specifically, the province of Alberta. Our estimate for the HBL, of 2.7 to 3.4 Tg y$^{-1}$, is close to the prior estimate (which includes wetland fluxes from the land surface model, LPX-Bern) and to other independent inversion estimates. However, our estimate for Alberta, of 5.0 to 5.8 Tg y$^{-1}$ is significantly higher than

the prior (which also includes anthropogenic sources from the EDGAR-4.2FT2010 inventory). Since the fluxes from this region persist throughout the winter, this may signify that the anthropogenic emissions are underestimated. For North Eurasia, we find a mean source of 52.2 to 55.5 Tg y$^{-1}$, with a strong contribution from fluxes in the Western Siberian Lowlands





(WSL) for which we estimate a source of 19.3 to 19.9 Tg $y^{-1}$. Over the 9-year inversion period, we find significant year-to-year variations in the fluxes, which in North America and, specifically, in the HBL appear to be driven at least in part by soil temperature, while in the WSL, the variability is more dependent on soil moisture. Moreover, we find significant

positive trends in the $CH_4$ fluxes in North America of 0.38 to 0.57 Tg $y^{-2}$, and North Eurasia of 0.76 to 1.09 Tg $y^{-2}$. In North America, this could be due to an increase in soil temperature, while in North Eurasia, specifically, Russia, the trend is likely due, at least in part, to an increase in anthropogenic sources.

**Keywords**: methane, emissions, wetlands, atmospheric transport modelling, atmospheric inversion, Arctic, boreal.

## 1. Introduction

Atmospheric methane ($CH_4$) increased globally during the 20[th] century, from a pre-industrial

value of approximately 722 ppb (Etheridge et al. 1998) to 1773 ppb in 1999 (Kirschke et al. 2013), largely due to an increase in anthropogenic sources. The upward trend was interrupted between 1999 and 2006, when the atmospheric growth rate of $CH_4$ was close to zero but resumed again around 2007 (Dlugokencky et al. 2011; Kirschke et al. 2013; Nisbet et al. 2014) and, in 2014, the growth rate was substantially faster (11.5 ppb $y^{-1}$) than in any other

20 year since 2007 (http://www.esrl.noaa.gov/gmd/aggi/aggi.html). Changes in the atmospheric growth rate indicate changes in the balance of $CH_4$ sources and sinks, however, the cause of the 1999-2006 stabilization and subsequent rise in atmospheric $CH_4$, and its attribution to different sources is still not fully resolved (Nisbet et al. 2014).

The main $CH_4$ sources are either biogenic such as wetlands, rice-paddies, landfills and enteric fermentation, thermogenic such as fossil fuels, geological seeps and mud volcanoes or pyrogenic such as the combustion of fossil and bio fuel and biomass (Kirschke et al. 2013, and references therein). Different combinations of sources, particularly, wetlands, fossil fuels, and enteric fermentation have been proposed as the reason for the rise of atmospheric $CH_4$

since 2007 (Kirschke et al. 2013; Schaefer et al. 2016). Studies using atmospheric observations in global inversion models have pointed to an increase in tropical wetland emissions as well as in anthropogenic sources, especially fossil fuels, in the temperate northern hemisphere after 2006 (Bruhwiler et al. 2014; Bergamaschi et al. 2013). However, anomalously high temperatures in the Arctic in 2007 are thought to have also resulted in



higher wetland emissions and, consequently, contributed to the high growth rate of $CH_4$ in the same year (Dlugokencky et al. 2009; Bousquet et al. 2011; Rigby et al. 2008).

In recent decades, the high latitudes have warmed substantially with temperatures in the Arctic increasing at an average rate of 0.38°C per decade (Chylek et al. 2009) and the changing climate may have a considerable impact on $CH_4$ sources (Bridgham et al. 2013; AMAP, 2015). The northern high latitudes contain about 44% of the world's natural wetlands (Lehner and Döll, 2004) and contribute about 24 Tg $y^{-1}$, equivalently 26%, to the global natural wetland source of $CH_4$ (Cao et al. 1998). Temperature can impact wetland sources of $CH_4$ directly, by influencing the metabolic rate of methanogens (Christensen et al. 2003; Christensen et al. 1996), and indirectly, via permafrost melt and changes to the water table depth, snow cover, and evapotranspiration (Bohn et al. 2007). Other climatic changes in the Arctic, such as increasing sea temperature and sea-ice loss (Stroeve et al. 2007) may also increase the source of $CH_4$ through possible destabilization of methane hydrates in ocean sediments (e.g. Shakhova et al. 2010; Biastoch et al. 2011). In addition to natural sources, the oil and gas industries are particularly important sources in the high northern latitudes, especially in Russia where 20% of the world's natural gas is produced. With an estimated leak rate of 1 to 10% of gas production (Hayhoe et al. 2002) this would release 3.5 to 35 Tg $y^{-1}$ of $CH_4$. Given the rapid rate of climate change in the high latitudes and possible future expansion of anthropogenic activities in the Arctic it is imperative to have accurate estimates of present-day $CH_4$ emissions, to better understand their natural variability, and to establish any trend.

Atmospheric observations can place a mass balance constraint on emissions and, since they are available quasi-continuously over time scales of several years, can be used to examine inter-annual variations and trends. This method is formalized in atmospheric inversions in which fluxes are found by minimizing a cost function that includes the comparison of the observations and mixing ratios modeled from a prior estimate of the fluxes using a model of atmospheric transport (e.g. Enting, 2002). Until relatively recently, however, atmospheric constraints on $CH_4$ emissions in the high northern latitudes were limited by the scarcity of observations in this region. Until the early 2000s, the ground-based network north of 50°N was limited to about ten discrete sampling sites and three sites with quasi-continuous monitoring (i.e., Barrow in Alaska, and Alert and Fraserdale in Canada). Satellite data in this region are also limited. Satellite measurements of $CH_4$ are made using either thermal infrared



(TIR) sounders, such as IASI, or near-infrared (NIR) measurements, such as by the SCIAMACHY instrument (onboard ENVISAT, 2003 – 2012) and the TANSO instrument (onboard GOSAT, since 2009). However, NIR measurements are limited to the summer and the availability of data is further reduced owing to filtering for aerosols and cloud cover. On

5   the other hand, TIR measurements from IASI have lower sensitivity to near surface concentrations compared to TANSO or SCIAMACHY and have particularly large biases in the high latitudes (Xiong et al. 2013). Therefore, satellite observations of $CH_4$ from latitudes higher than 50° are presently not included in global atmospheric inversions (e.g. Bergamaschi et al. 2013; Houweling et al. 2014; Alexe et al. 2015; Monteil et al. 2013).

The observation coverage, however, has recently improved with the establishment of JR-STATION (Japan-Russia Siberian Tall Tower Inland Observation Network) (Sasakawa et al. 2010) and the Zotino Tall Tower Observatory (ZOTTO) (Winderlich 2010) in Siberia, the Pallas station in Finland (Aalto et al. 2007), and the Environment Canada (EC) network in

Canada (Worthy et al. 2003) (see Fig. 1). Some of the data from these networks/sites have been included in previous inversion studies, however, only with limited spatial and temporal coverage: the JR-STATION observations have been used in an inversion focusing on the Siberian lowlands in the year 2010 (Berchet et al. 2015) and the EC observations have been used in an inversion focusing on Canada and the US for 2007 and 2008 (Miller et al. 2014;

Miller et al. 2016). We combine observations from all these new networks and stations in an atmospheric inversion for $CH_4$ focusing on the entire region north of 50°N, and over the period 2005 to 2013, when most observations are available. With the observational constraint of these data, a more accurate estimate of high latitude fluxes and their inter-annual variability is possible compared to earlier estimates, for which much of these data were not available (e.g.

Bruhwiler et al. 2014; Bergamaschi et al. 2013).

In section 2, we present an overview of the inversion method and a description of its various components, including the observations, the atmospheric transport modelling, and the a priori fluxes. In Section 3, we analyse the performance of the inversions and compare the results to

independent observations, i.e., data not used in the inversion. Section 4 discusses the spatial and temporal variability of the a posteriori fluxes and evidence for trends.

## 2. Methodology

### 2.1 Overview of the inversion framework





In this study we use the Bayesian inversion framework, FLEXINVERT, which is based on the Lagrangian particle dispersion model, FLEXPART (Stohl 1998; Stohl et al. 2005), and is fully described in Thompson and Stohl (2014) and has been previously used for $CH_4$ inversions (Thompson et al. 2015). FLEXINVERT uses so-called flux sensitivities (or source-

receptor relationships) calculated by FLEXPART, which describe the relationship between a flux over a given area and time interval, and the associated change in atmospheric mixing ratio (in this case of $CH_4$) at another time and location (see section 2.2). FLEXINVERT finds the optimal $CH_4$ fluxes by minimizing the mismatch between modelled and observed mixing ratios with a constraint imposed by prior information and its uncertainty. This is described by

the cost function:

$$J(\mathbf{x}) = \frac{1}{2}(\mathbf{x} - \mathbf{x_b})^{\mathrm{T}}\mathbf{B}^{-1}(\mathbf{x} - \mathbf{x_b}) + \frac{1}{2}(\mathbf{y} - \mathbf{Hx})^{\mathrm{T}}\mathbf{R}^{-1}(\mathbf{y} - \mathbf{Hx}) \tag{1}$$

where $\mathbf{x_b}$ and $\mathbf{x}$ are respectively the prior and posterior state vectors of surface fluxes and $\mathbf{H}$ is a Jacobian matrix of flux sensitivities, which relates the fluxes to the observed mixing ratios, $\mathbf{y}$. The observation-model and posterior-prior flux mismatches are weighted by their

uncertainties, described by the error covariance matrices, $\mathbf{R}$ and $\mathbf{B}$, respectively. The optimized state vector is found by solving the first order derivative of the cost function:

$$J'(\mathbf{x}) = \mathbf{B}^{-1}(\mathbf{x} - \mathbf{x_b}) - \mathbf{H}^{\mathrm{T}}\mathbf{R}^{-1}(\mathbf{y} - \mathbf{Hx}) = 0 \tag{2}$$

FLEXINVERT takes advantage of the fact that in this case $\mathbf{H}$ is a matrix operator and solves Eq. 2 directly (i.e., analytically).

The second order derivative of the cost function is equal to the reciprocal of the posterior uncertainty. Although, this could be used to calculate the posterior uncertainty, we find for our inversion framework that this leads to an underestimate of the uncertainty. This is because the matrix $\mathbf{B}$ is not stored in computer memory but rather its Eigen decomposition with

truncation of the smallest 0.01% of eigenvalues and their corresponding eigenvectors. For this reason, we use a Monte Carlo approach to calculate the posterior uncertainty following Chevallier et al. (2005).

We optimize the fluxes at monthly temporal resolution and on a grid of varying spatial

resolution (with a finest resolution of 1°×1°, see section 2.4 and Fig. 2) over the period 2005 to 2013. The various components of the inversion framework are described in the following sections.





### 2.2 Atmospheric observations

In this study, we make use of relatively new stations measuring $CH_4$ dry-air mole fractions (units of nmol mol$^{-1}$ or parts per billion, ppb) established in Siberia, Europe, and Canada (see Fig. 1 and Table 1). The JR-STATION network was established in 2004 and has expanded to

a total of 9 tall tower sites with in-situ measurements of $CH_4$ (Sasakawa et al. 2010). Measurements of $CH_4$ are made using a semi-conductor sensor based on a tin dioxide natural gas leak detector that was developed for atmospheric measurements by Suto and Inoue (2010). The measurements have a repeatability of 3 ppb and are reported on the NIES-94 scale. The tall towers have two sampling inlets at different heights, each of which is sampled for 20

minutes during a 1-hour period (Sasakawa et al. 2010). In this study we use observations from seven towers and only from the upper inlets, which have heights between 43 and 85 metres above ground level (magl). Three towers (KRS, DEM, and NOY) are surrounded by taiga forest and wetlands. The IGR tower is located in a small town (population ~10,000) by the Ob River and is also surrounded by wetlands, the AZV and VGN towers are located in steppe

regions, and the YAK tower is located in the East Siberian Taiga (for details see Sasakawa et al. 2010).

EC established two in-situ measurement sites more than 20 years ago, ALT and FSD, but since the early 2000s the network has expanded considerably. In this study, we include 6 sites,

which have records of at least 4 years. Throughout most of this study, the measurements were made with approximately hourly resolution using Gas Chromatographs equipped with Flame Ionisation Detectors (GC-FID) and are reported on the WMOX2004 scale. However, at LLB and FSD the GC-FID instruments were replaced by Cavity Ring-Down Spectroscopy (CRDS) instruments in 2012 and 2013, respectively. FSD is located on the southern perimeter of the

Hudson Bay Lowland and is surrounded by boreal forest and the closest town is approximately 70 km away (population of ~2500). CDL is in the province of Saskatchewan and is also surrounded by boreal forest and the closest town is 100 km away (population of ~35,000). ETL was established to replace CDL, which was discontinued in 2008. ETL is located 40 km away from CDL at the site of a 106 m high communication tower (compared to

the 28 m high tower at CDL) and is also surrounded by boreal forest. LLB is located in a region of peatland and boreal forest in the province of Alberta and is approximately 200 km away from the city of Edmonton (population ~800,000). CHM is located in a boreal forest in eastern Canada in a cool humid climate. CHL is located on the southern coast of the Hudson

Bay with boreal forest to the south and Arctic tundra to the northwest. Flask-air sampling began at CHL in May 2007 and in-situ measurements in October 2011.

The Pallas station is located in northern Finland in an area of wetlands with some lakes and patches of forest (Aalto et al. 2007). Continuous $CH_4$ measurements have been made from 2004 to 2009 using a GC-FID instrument and, since 2009, using a CRDS instrument. The data are reported on the WMOX2004 scale and have an average repeatability of 1.5 ppb. Air is sampled from an inlet at 7 magl.

ZOTTO (http://www.zottoproject.org/) is a tall tower site surrounded by light taiga forest interspersed with bogs and has a $CH_4$ time series available since 2009 (Winderlich 2010). The measurements are made using a CRDS instrument and buffer volumes on each sample line, which allows a continuous, near-concurrent measurement of air from six heights. For this study, we use data from the uppermost level at 301 magl. Data from each height are reported at circa 20-minute intervals on the WMOX2004 scale with a repeatability of 0.3 ppb (Winderlich 2010).

The Zeppelin observatory is located on a mountaintop on the island of Spitsbergen (the largest island of Svalbard) and has a $CH_4$ record dating from 2001. Methane measurements were made using a GC-FID until April 2012 and were reported hourly with a repeatability of 3 ppb (from 2004 to 2012). In April 2012, the GC-FID was replaced by a CRDS instrument and data since then are reported with a repeatability of 0.2 ppb at 1-minute time resolution. Up until spring 2011, air was sampled from 2 m above the roof at the Zeppelin observatory and from then on, from a 15 m-tall mast. The whole time series from August 2001 has been reprocessed as a part of the harmonisation of historic concentration measurements within the European Commission FP7 project, InGOS (http://www.ingos-infrastructure.eu/).

The NOAA flask-air sampling network includes eight sites north of 50°N (Dlugokencky et al. 2015). In this study, we use five of these sites that sample predominantly background air to define the initial mixing ratios (see section 2.5) and the remaining three sites (BAL, CBA, and TIK) are used in the inversion. Discrete samples in the NOAA network are taken at approximately weekly frequency and are analysed in a central laboratory using a GC-FID. The concentrations are reported on the WMOX2004 scale and have a long-term repeatability of 2 ppb.





All data were adjusted to the WMOX2004 scale using the results of the World Meteorological Organisation Round Robin Comparison Experiment (http://www.esrl.noaa.gov/gmd/ccgg/wmorr/). We assimilated the in-situ observations as daily afternoon (between 12:00 and 16:00 local time) averages and the discrete observations without data selection or averaging as these samples are normally taken during daytime when the boundary layer is well mixed. Further data selection is performed on the in-situ observations and is discussed in section 2.6.

## 2.3 Prior information

For wetland fluxes, we compare estimates from two different land surface models. The first is the LPX-Bern model (Land Surface Processes and Exchanges model of the University of Bern) (Spahni et al. 2013). LPX-Bern calculates $CH_4$ fluxes monthly at 1°×1° resolution for peatlands, inundated wetlands, wet and dry mineral soils, and rice paddies. The wetland fraction and water table depth in each grid cell are calculated dynamically using the DYPTOP model (DYnamical Peatland model based on TOPmodel) (Stocker et al. 2014). The second model is LPJ-DGVM (Lund-Potsdam-Jena Dynamic Global Vegetation Model; Bergamaschi et al. 2007). LPJ-DGVM uses a fixed wetland area based on land-cover maps (Bergamaschi et al. 2007). In LPJ-DGVM substrate availability for methanogenesis is represented as the total soil respiration rate, which integrates the fast-turnover soil organic matter pool and soil temperature, and the water table depth is based on soil moisture (Bergamaschi et al. 2007). LPJ-DGVM calculates monthly $CH_4$ fluxes at 1°×1° resolution for inundated wetlands, wet soils and peatlands, but does not calculate the uptake of $CH_4$ by dry soils or the emissions from rice paddies. For the dry soil flux of $CH_4$, we use the monthly climatology of Ridgwell et al. (1999) (also at 1°×1° resolution) and for rice paddies, we use the estimate from the Emissions Database for Global Atmospheric Research (EDGAR-v4.2 FT2010, from hereon, EDGAR) with monthly emission factors from Matthews et al. (1991).

For the anthropogenic sources, which include emissions from enteric fermentation in domestic animals, oil, gas and coal exploitation, and landfills, we use annual estimates from EDGAR, which are provided at 0.1°×0.1° resolution. Since EDGAR is only available up to 2010, we used the 2010 emissions also for the years 2011 to 2013. Estimates of emissions from biomass burning were taken from the Global Fire Emissions Database (GFED-v3) (van der Werf et al. 2010), which are provided at monthly and at 0.5°×0.5° resolution. For ocean





fluxes of $CH_4$, we use the monthly climatology at 1°×1° resolution from Lambert and Schmidt (1993) and for the emissions from wild ruminant animals and termites, we use the monthly climatologies of Houweling et al. (1999) and Sanderson et al. (1996), respectively, both at 1°×1° resolution. The different soil flux estimates, i.e., LPX-Bern and LPJ-DGVM,

form the basis of two different prior fluxes and, correspondingly, inversion scenarios, S1 and S2 (for an overview of the prior fluxes, see Table 2).

Prior flux uncertainties were calculated as 50% of the flux in each grid cell with minimum and maximum values of $1×10^{-12}$ and $1×10^{-6}$ kg m$^{-2}$ h$^{-1}$, respectively. The prior flux error

covariance matrix, **B**, was calculated as the Kronecker product of the temporal and spatial correlation matrices, which were calculated using an exponential decay model and a temporal scale length of 90 days and spatial scale lengths of 500 km over land and 2000 km over ocean. The square root of the sum of all elements in the error covariance matrix was scaled to 15 Tg y$^{-1}$, representing ~25% of the total area integrated flux in the domain.

### 2.4 Flux sensitivities

The flux sensitivities are calculated using the backwards mode of FLEXPART, in which virtual particles are followed backwards in time from the observation times and locations (from hereon referred to as "receptors"). At 3-hourly intervals, 10000 virtual particles were

released and followed backwards in time for 10 days to any point on the globe. Flux sensitivities are calculated as proportional to the average residence time of $P$ back-trajectories in a given grid cell and time step (i, n):

$$\frac{\partial y}{\partial x} = \frac{1}{P}\sum_{p=1}^{P}\frac{\Delta t'_{i,p,n}}{\rho_{p,n}} \tag{3}$$

where $\rho_{i,n}$ is the air density and $\Delta t'_{i,p,n}$ is the residence time of trajectory $p$ in the spatio-

temporal grid cell (i, n) (for details see Seibert and Frank 2004). FLEXPART simulations were made using European Centre for Medium-Range Weather Forecasts ERA-Interim (ECMWF EI) meteorological analyses with a 12-hour analysis window and a spatial resolution of approximately 80 km (T255) on 60 vertical levels. The analyses were gridded to 1.0° × 1.0° and interpolated to 3-hourly fields. EI was chosen over the higher resolution

operational data because of its long-term consistency. Loss of $CH_4$ due to oxidation by OH radicals was also calculated along the trajectories using pre-calculated OH fields from the GEOS-Chem model. However, over the 10-day calculation period the loss is generally small, i.e., ~1 ppb.





The average flux sensitivity of all receptors over the year 2009 is used to define a grid of varying spatial resolution for the inversion (the grid was kept constant throughout the 9-year inversion). This grid has maximum resolution of $1.0° \times 1.0°$ in regions where there is a strong

contribution from fluxes to the change in $CH_4$ mixing ratios at the measurement sites and decreasing resolution in steps of factor two up to $8.0° \times 8.0°$ in regions where there is only a weak contribution (Thompson and Stohl 2014; Stohl et al. 2010) (Fig 2.). In this way, the number of state variables is reduced without significantly increasing the aggregation error.

The flux sensitivities are calculated globally but only those north of 50°N are used to optimize the fluxes. Sensitivities to surface fluxes outside this domain are generally low; however, we calculate their contribution to the change in mixing ratio at each observation time and location by integrating the product of flux sensitivity and prior fluxes outside the domain, following Thompson and Stohl (2014). This is in addition to the contribution from mixing ratios at the

termination points of the virtual particles, i.e. 10 days before the observation was made (the definition of these *initial* mixing ratios is discussed in section 2.5).

### 2.5 Initial and background mixing ratios

Since the Lagrangian simulations only account for changes in mixing ratios due to fluxes up

to 10 days before an observation was made, we need to provide an estimate for the contribution prior to that, i.e., the *background* mixing ratio. The background mixing ratios were calculated by coupling FLEXPART to *initial* mixing ratio fields according to the method of Thompson and Stohl (2014). Initial mixing ratio fields were calculated for the global troposphere, at monthly temporal and 10° longitude by 5° latitude spatial resolution,

using a bivariate interpolation of flask-air samples in the NOAA network (Fig. 3a.). We used flask-air samples with approximately weekly frequency from 98 sites globally (note that none of the sites used to determine the initial mixing ratios were used in the inversion). Before being used in the interpolation, the time series from each site was filtered for short-lived signals (of less than one week) using a local regression method (Ruckstuhl et al. 2012) and

gaps of more than one month were filled using a four harmonic plus second order polynomial function fitted to the whole time series. For the lower and mid stratosphere, we used mixing ratio fields from the TM5 model (Bergamaschi et al. 2015).

The contribution of the initial mixing ratios to the mixing ratio at a given observation time and location, i.e., the *background* mixing ratio, was calculated using the FLEXPART back-trajectories. The background mixing ratio is equal to the weighted average of the initial mixing ratios in the grid cells where the back-trajectories terminated 10 days prior to the observation. Figure 3b shows an example of the sensitivity to the initial mixing ratios calculated for the site IGR for January 2009. The background mixing ratio at each site is shown in the Supplementary Information Fig. 1.

### 2.6 Data selection and observation uncertainties

Many of the sites used in the inversion are located in the interior of a continent and at high latitude. At such locations, low wind speeds and strong surface-based temperature inversions can occur in winter. Under these conditions, $CH_4$ from local sources can accumulate in shallow layers and atmospheric mixing ratios are extremely difficult to model, as they are very sensitive to errors in the planetary boundary layer (PBL) height and PBL mixing. To examine the impact of the shallow PBL on the simulated mixing ratios, we performed a test in FLEXPART in which the condition of a minimum PBL height of 100 m was removed (see Fig. 4). Interestingly, the simulations with no set minimum PBL height were very similar to those with a minimum of 100 m, indicating that the PBL height based on the meteorological reanalyses is greater than 100 m most of the time.

As the model representation errors for observations made during these very cold periods are very large and not normally distributed, we have applied data selection criteria to avoid assimilating these observations in the inversion. At all sites in the JR-STATION network, temperature is measured at two heights, with the lower level between 11 and 45 magl and the upper between 43 and 85 magl, and wind speed is measured at one height. We filtered observations for times when the vertical temperature gradient from the lower to upper level was positive, i.e., the upper level was warmer by at least 1°C, indicating atmospheric inversion conditions, and when the wind speed was below 3 ms$^{-1}$ (see Fig. 4). By applying these selection criteria, periods when very large positive excursions (of several hundred ppb) were observed, but not captured by the model, were removed.

In the case of IGR, the model representation errors are likely the largest of all sites owing to the relatively low intake height of 47 magl and its location in a small town, meaning that there are emissions in its near field, which cannot be resolved in the model. For other continental





sites outside the JR-STATION network the temperature was not available at two heights so this criterion could not be used. However, the estimated model representation errors are likely much smaller at these sites (see Supplementary Information Fig. 1). In the case of ZOT, this is due to the height of the air intake, at 301 magl, compared to only 43 to 85 magl in the JR-STATION network, while PAL is located on a hill (565 masl), and both sites are remote from towns or industry. The continental sites in the EC network, FSD, CDL, LLB and ETL, are also remote from major towns or industry.

Uncertainties in the observation space included estimates for the uncertainty in the measurements and model representation. For the measurement uncertainty, we used a conservative estimate of 5 ppb, which is larger than the precision stated by the data providers to account for variations in the instrumental performance. For the model representation uncertainty we included estimates of the transport uncertainty for the transport within the inversion domain and for the background mixing ratios. These estimates were based on differences between FLEXPART simulations made using different meteorological reanalysis data, i.e., ECMWF EI versus NCEP FNL. Since it is computationally demanding to run all FLEXPART calculations twice, we only ran simulations with the two meteorological datasets for 2009 and calculated the mean daytime (12:00 to 16:00) errors for each site and month. This proxy can be considered a lower bound for the true transport uncertainties but provides an indication of the magnitude and spatiotemporal variability of these uncertainties. We estimated uncertainties for the transport within the domain of 2 to 65 ppb and for background mixing ratios of 2 to 22 ppb. The uncertainty is larger for continental sites and in the winter months as expected due to the challenges of representing the shallow PBL (see Supplementary Information Fig. 1 and 2). In addition to the transport uncertainties, we included an estimate of the temporal representation uncertainty that arises due to the averaging of observations to a daytime mean. This last uncertainty was estimated simply as one standard deviation of the daytime observations. The total uncertainty in the observation space was calculated as the square root of the quadratic sum of the measurement uncertainty and the three model representation uncertainties. We used the square of the observation uncertainty for the diagonal elements of the observation error covariance matrix and assumed the off-diagonal elements to be zero, i.e., that there is no error correlation between observations. As we use mean daytime observations, this assumption is reasonable.

## 3. Results





### 3.1 Comparison of simulated and observed mixing ratios

We start with the comparison of the simulated and observed $CH_4$ mixing ratios, specifically, the comparison between simulations using different prior flux estimates. Figure 5 shows Taylor diagrams of the Pearson's correlation coefficient and normalized standard deviation
(NSD) for scenarios using the two different prior estimates, i.e., S1 and S2, at each site. It is noteworthy that the two prior estimates are distinguishable at most sites signifying that the observations are sensitive to differences between these. The correlation, a priori, is mostly higher than 0.3 except at the sites CHM and ESP, which have quite low variability and sensitivity to fluxes in the domain (see Supplementary Information Fig. 1a). In general, the
NSD for both prior flux estimates is significantly less than one, indicating that the variability in the mixing ratios is underestimated. It is to some extent expected that NSD is underestimated owing to the limited spatiotemporal resolution of the fluxes; however, it is especially apparent for continental sites, e.g., LLB, ETL, IGR, KRS, YAK and ZOT. Supplementary Fig. 1 shows that the prior fluxes not only underestimate the variability at
these sites but also that the prior mixing ratios have a low bias. Overall, the prior flux estimate in S2 (which include the LPJ-DGVM wetland emissions) gives a slightly closer fit to the observations as seen from the lower cost (see Table 3).

A posteriori, the simulated mixing ratios from inversion scenarios S1 and S2 are almost
indistinguishable and the difference in the cost between these inversions is small. Furthermore, the mixing ratio simulated with the posterior fluxes matches the observations significantly better than a priori, as expected, and the posterior observation-model mismatches are nearly normally distributed and mostly fall within the assumed uncertainty range of the observations (see Supplementary Information Fig. 3). The comparison of the fit to the observations a
posteriori is not a sufficient indicator of the inversion performance as the closeness of the fit a posteriori depends on the weighting given to the observation-model and prior-posterior flux mismatches (see Eq. 1). A better indicator of the inversion performance is the comparison of the posterior simulated concentrations with observations that were not included in the inversion, i.e., independent observations. Figure 6 shows the comparison of the prior and
posterior simulations from scenario S1 with observations from regular aircraft profiles at Poker Flats (PFA, 65.1°N, 147.3°W), Estevan Point (ESP, 49.4°N, 126.5°W) and East Trout Lake (ETL, 54.3°, 105°W). The comparison was made for three altitude levels, from 4 to 10 km, 1 to 4 km, and 0 to 1 km. Ground-based measurements at ESP and ETL were included in the inversion, therefore, the aircraft data at the lower level are not completely independent.



Most notable is that the prior fluxes result in a low bias compared to observations at ETL throughout the inversion period, which is visible up to ~4 km (Fig. 6a). Using the posterior fluxes, this bias is reduced and the correlation increased from 0.33, a priori, to 0.37, a posteriori (Fig. 6b). At the other independent sites, PFA and ESP, there is only a modest
improvement a posteriori versus a priori.

### 3.2 Comparison of the posterior fluxes from different scenarios

In addition to the inversion scenarios S1 and S2, using different prior flux estimates, we included a scenario, S3, to test for the impact of the changing observation network with time.
Scenario S3 uses the same prior fluxes as S1 but includes only sites with quasi-continuous records over the period of inversion (see Table 1 and Supplementary Information Fig. 4). Figure 7 shows the annual mean posterior fluxes, as well as the posterior-prior flux differences, obtained for scenarios S1, S2 and S3 (note we show only the result for 2009 as the other years were analogous). For all three scenarios, the posterior flux distribution is very
similar with the largest fluxes in the vicinity of the Western Siberian Lowlands (WSL), Europe, western Canada, and around the Hudson Bay Lowlands (HBL). In addition, there are a number of hot spots, notably in Eastern Europe and Russia, which are anthropogenic emissions present in the EDGAR inventory and are largely unaltered by the inversions. The distribution of the flux increments (i.e., the posterior-prior flux difference) is also similar
across the three scenarios. This is not unexpected for S1 and S3 (Fig. 7b and 7f) as they use the same prior estimates, but it is noteworthy that S2 shows the same pattern of positive increments in the WSL and Eastern Canada and negative increments in Europe and western Russia. One difference in S2 compared to S1 and S3 though, is the negative increment in the HBL area, which results from the higher prior estimate of the LPJ-DGVM model for wetland
fluxes in this region (S1 and S3, which use the LPX-Bern wetland fluxes, show almost no change for the HBL area).

Figure 8 shows the uncertainty reduction for scenarios S1 and S3 calculated as $1 - \sigma_{post}/\sigma_{prior}$ where $\sigma$ is the uncertainty in each grid cell. Since S2 uses the same
observations and prior uncertainty estimates as S1, the uncertainty reduction is equal to that of S1 and is, therefore, not shown. Scenarios S1 and S2 include 19 sites and a total of 3291 observations in 2009, compared to 12 sites and 2499 observations in S3. The impact of the additional sites of YAK, VGN, AZV and ZOT in S1 and S2 can be seen in the greater uncertainty reduction in Siberia, and that of the site CHL, in Canada. In all scenarios,



however, the uncertainty reduction is greatest in Europe, western Siberia, and Canada but modest in eastern Siberia, Greenland and Alaska.

Figure 9 shows the area-integrated fluxes for northern North America (Canada and Alaska) and North Eurasia (Europe and Russia) from the three scenarios. For North America, there is a clear difference between the prior fluxes in the phase of the seasonal cycle and the summer maximum, which is due to the different models used for wetlands (i.e., LPX-Bern versus LPJ-DGVM). In contrast, there is no significant difference between the posterior fluxes from scenarios S1 to S3, indicating first, that the seasonal cycle is well constrained in the inversion, and second, that the discontinuity of observations from the sites CHL and CHM has little impact on the northern North American integrated fluxes. For North Eurasia, there are again differences in the prior fluxes, owing to the choice of land surface model, which are no longer visible in the posterior fluxes. While the results from scenarios S1 and S2 are very similar, S3 results in higher summer maxima from 2008 onwards. This is due to the exclusion of data from four sites (YAK, VGN, AZV and ZOT) in S3, which means the inversion is more dependent on the remaining observations at IGR, DEM, and KRS for constraining fluxes in Siberia. A corollary of this is that since only these sites are available up to 2007, the posterior fluxes may be overestimated for 2005 to 2007 also in scenarios S1 and S2. The temporal variability of the fluxes is discussed in more detail in section 4.2.

## 4. Discussion

High latitude (>50°N) $CH_4$ emissions are largely from anthropogenic sources (particularly oil and gas exploitation), ~60%, and natural wetlands, ~40%, according to the prior estimates used in this study. The different sources are not always spatially distinct at the resolution of the emission sensitivities (in this study 1°×1°). In this case, it is not possible to resolve them without the use of additional atmospheric tracers, such as changes in the $^{13}C$ to $^{12}C$ isotope ratio in $CH_4$ (written as $\delta^{13}C$), which is sensitive to the emission source (e.g. Dlugokencky et al. 2011). Currently observations of $\delta^{13}C$ are much scarcer than for $CH_4$ mixing ratios and we have not included these in our inversion. Therefore, we focus the discussion mainly on the distribution of the total $CH_4$ source.

We present regional emission totals giving the range of scenarios S1 and S2, which are better constrained than S3 as they include all available observations. The range is generally close to the uncertainty calculated for each scenario and which is given in Table 4. We, however,



include S3 in the analysis of the inter-annual variability and trends, as this scenario uses only observations from sites that are available for most of the inversion period, thus year-to-year differences are independent of changes in the observation coverage.

## 4.1 Spatial distribution of the fluxes

### 4.1.1 North America

For northern North America (>50°N), we estimate a mean total source of 16.6 to 17. 9 Tg y$^{-1}$ for 2005 – 2010 (the period overlapping with independent global inversion studies), which is 7.1 and 3.7 Tg y$^{-1}$ higher than the prior estimates in scenarios S1 and S2 (which included wetland flux estimates from LPX-Bern and LPJ-DGVM, respectively). Our estimate is also substantially higher than the global inversion estimates of Carbon-Tracker $CH_4$, of 8.1 Tg y$^{-1}$ (Bruhwiler et al. 2014), and TM5, of 14.0 Tg y$^{-1}$ (Bergamaschi et al. 2013) (see Table 4). However, it is lower than the regional inversion estimate of Miller et al. (2014) for Canada, of 21.3 ± 1.6 Tg y$^{-1}$ for 2007 – 2008. The regions dominating northern North American $CH_4$ fluxes, and where we see differences from the prior estimates, are the HBL and western Canada.

Methane fluxes in the HBL (50-60°N, 75-96°W) are largely dominated by wetlands, more than 90% of the total according to the prior estimates (see Supplementary Information Fig. 5). For the HBL, the posterior fluxes from scenarios S1 and S2 are 2.7 to 3.4 Tg y$^{-1}$ and are close to the prior estimate of S1, of 2.7 Tg y$^{-1}$, and to the inversion estimates of Miller et al. (2014), of 2.4 ± 0.32 Tg y$^{-1}$, Bruhwiler et al. (2014), of 2.7 Tg y$^{-1}$, and Pickett-Heaps et al. (2011), of 2.3 ± 0.3 Tg y$^{-1}$. However, our posterior fluxes are significantly lower (by 1.0 to 1.7 Tg y$^{-1}$) than the prior estimate of S2, which included wetland fluxes from LPJ-DGVM. Land surface models vary greatly in the estimate for HBL wetland emissions, from 2.2 to 11.3 Tg y$^{-1}$, due at least partly to the definition of wetland extent (Melton et al. 2013). The large discrepancy in LPJ-DGVM could be due to an over simplified calculation of water table depth, for which soil moisture is used as a proxy. LPX-Bern, in comparison, uses a dynamical calculation of water table depth based on the DYPTOP model.

In western Canada, the main source region approximately corresponds to the province of Alberta (in this study we compare the region 50-60°N, 110-120°W). In the prior flux estimates in scenarios S1 and S2, this region has quite low $CH_4$ emissions, 1.6 and 3.0 Tg y$^{-1}$, respectively. In contrast, all scenarios find a much larger source a posteriori, of 5.0 to





5.8 Tg y$^{-1}$, with most of the increase in the southern part of the province where there are no significant wetlands. Miller et al. (2014) also found large fluxes in Alberta and, similar to their study, we find that these fluxes persist throughout the year (Fig. 10 and 11) unlike fluxes dominated by wetlands, which have a strong seasonal cycle (see the HBL region in Fig. 11),

suggesting that they may be of anthropogenic origin. We hypothesise that the emissions are largely from natural gas production since Alberta produces 72% of Canada's natural gas and has Canada's largest shale gas reserves (https://www.neb-one.gc.ca/index-eng.html). Assuming that the wintertime wetland emissions are negligible, and that the seasonal variation in anthropogenic emissions is small (which has been found for emissions from oil

and gas operations in North America (Smith et al. 2015)), we use the wintertime emissions as an estimate for the anthropogenic source. Our estimate of $4.3 \pm 1.3$ Tg y$^{-1}$ is approximately 3 times larger than that of EDGAR and, for comparison, approximately 8 times larger than the estimate of the GAINS model (Höglund-Isaksson 2012), suggesting that anthropogenic sources in Alberta are currently strongly underestimated in at least these two inventories.

### 4.1.2 North Eurasia

For North Eurasia, we estimate mean total emissions of 52.5 to 55.5 Tg y$^{-1}$ (scenarios S1 and S2) for 2005 – 2010, which is 8.9 to 10.8 Tg y$^{-1}$ higher than the prior estimates. Our estimates are close to that found by the Carbon-Tracker $CH_4$ inversion of 49.7 Tg y$^{-1}$ (Bruhwiler et al.

2014) but much larger than those found by the TM5 inversions of approximately 34.0 Tg y$^{-1}$ (Bergamaschi et al. 2013) (see Table 4). One reason for the low estimate from the inversion of Bergamaschi et al. (2013) could be the poor observational constraint for the high northern latitudes; the inversion included only seven sites north of 50°N, all of which have flask-air samples at only approximately weekly intervals, and none of the sites were located in Siberia.

Also, their global inversion that included SCIAMACHY observations did not contain any satellite observations north of 50°N.

The main increase in the posterior fluxes, relative to both priors, is in the WSL region (50-75°N, 60-95°E). A posteriori, we estimate a source of 19.3 to 19.9 Tg y$^{-1}$, which is 7.1 to 8.9

30     Tg y$^{-1}$ higher than the prior estimates in scenarios S1 and S2, respectively. Our posterior estimates, however, fall in the mid-range of a recent study, which gave a tolerance interval of 5 to 28 Tg y$^{-1}$ for the total source in 2010 based on a regional inversion (Berchet et al. 2015). Wetlands cover 25% of the land surface in the WSL and are an important source of $CH_4$ in this region (Bohn et al. 2015). In addition, there is extensive oil and gas production in the




WSL, which is also a significant source of $CH_4$ and complicates the attribution of emissions from inversions to anthropogenic versus natural sources. Averaging the mean monthly emissions (from scenarios S1 and S2) for December to February, we estimate an anthropogenic source of $12.7 \pm 3.6$ Tg y$^{-1}$ or, equivalently, ~65% of the annual total. From the

5 residual, we estimate a wetland source of $6.9 \pm 3.6$ Tg y$^{-1}$, assuming that the summer wild fire emissions are very small, which based on the prior estimate of GFED-3.1, of ~0.08 Tg y$^{-1}$ is reasonable. We note, however, that anthropogenic sources in WSL may have some seasonal variation and that this would also affect the wetland source estimate. One such seasonally dependent source could be, for example, gas flaring, which has been suggested to have higher

$CH_4$ emissions in low temperatures owing to problems of igniting the flare. Our estimate of the wetland source is higher than the prior values from LPX-Bern and LPJ-DGVM of 4.9 and 5.9 Tg y$^{-1}$, respectively, but falls in the mid-range of estimates from Berchet et al. (2015), of 1 to 13 Tg y$^{-1}$, and is close to the mean of inversion estimates used in the inter-comparison study of Bohn et al. (2015), of $6.1 \pm 1.2$ Tg y$^{-1}$. On the other hand, it is lower than the

estimate of Bruhwiler et al. (2014) of 10.3 Tg y$^{-1}$. Our anthropogenic flux estimate is higher than that from EDGAR, of 6.8 Tg y$^{-1}$, and the inversion of Bruhwiler et al. (2014), of 8.1 Tg y$^{-1}$, but in the mid-range of estimates from Berchet et al. (2015), of 6 to 16 Tg y$^{-1}$ and lower than that of GAINS, of 19 Tg y$^{-1}$.

Large fluxes of $CH_4$ from the ocean to the atmosphere have been reported for the East Siberian Arctic Shelf (ESAS) with a source estimated to total 17 Tg y$^{-1}$ representing ~3% of the global source to the atmosphere (Shakhova et al. 2010; Shakhova et al. 2015). Although our inversion has only a modest reduction in uncertainty in the ESAS region (see Fig. 8) we do not find any evidence of a large source in this region. This is consistent with a recent study

based on atmospheric observations and inverse modelling, which found the ESAS region to be a source of only 0.5 to 4.5 Tg y$^{-1}$ (Berchet et al. 2016).

### 4.2 Temporal variability of the fluxes

### 4.2.1 Seasonal cycle

Emissions in the HBL region are dominated by wetlands. For this region, our inversion estimates a much later onset and more gradual increase in emissions in spring, as well as a later maximum, in August to September, compared to the LPJ-DGVM model, while these features of the seasonal cycle are better reproduced by the LPX-Bern model (Fig. 11). Both models, however, produce a more gradual decline in emissions following the summer




maximum compared to the inversion, which indicates a rapid decline from September to October and close to zero emissions from November to March. A similar pattern is seen for all of northern North America, suggesting that the wetland emissions are dominating the seasonality also at this scale.

One reason for the poor representation of the seasonal cycle in LPJ-DGVM could be that this model strongly emphasizes temperature control on $CH_4$ production in the high latitudes and may have a too strong correlation with soil temperature (Bergamaschi et al. 2007). Although the annual emission is largely dominated by fluxes during the growing season (Whalen et al.

1992), a few studies have indicated high fluxes during spring and fall due to thaw and freeze dynamics (Mastepanov et al. 2008; Zona et al. 2016; Sweeney et al. 2016). This process is also parameterized in the LPX-Bern model. Our inversion results, however, do not indicate any large-scale emissions in the HBL outside the growing season.

In the WSL region the maximum occurs in July to August, and is earlier than that of LPX-Bern in August and later than that of LPJ-DGVM in June. The maximum in July-August is consistent with the majority of land surface models in the inter-comparison study of Bohn et al. (2015). Similarly to the HBL, LPJ-DGVM produces a too early onset for the spring increase in emissions, again this may be due to an over sensitivity to temperature, and both

models have a more gradual decline in emissions in autumn compared to the inversion results. In contrast to the HBL, there are substantial emissions in winter, which are predominantly due to anthropogenic sources (Umezawa et al. 2012). Another notable feature is the small secondary peak in March. Berchet et al. (2015) also detected a March peak in 2010 and suggested that this may be due to anthropogenic emissions, in particular, from higher gas

production during the later winter. Although the mean seasonal cycle shows a peak in March, only five out of the nine years in our study had a spring peak, which occurred between March and April, and 2013 had a small peak in February. Some monthly variation in the anthropogenic emissions from oil and gas production is conceivable as very cold conditions can hamper production, which returns to normal levels only when temperatures start to

increase. Unfortunately, monthly production statistics from this region are not available to confirm this hypothesis. We also explored an alternative explanation for the spring peak, that is, wetland emissions during spring thaw, which have been previously observed in high latitudes from flux chamber and eddy covariance measurements (Tokida et al. 2007; Hargreaves et al. 2001; Mastepanov et al. 2013). Such emissions occur as snow melts and the



surface soil begins to thaw, allowing $CH_4$ that has been trapped below the surface during the winter, and which was formed by methanogenesis before the subsurface soil froze, to escape to the atmosphere. We examined air temperature measured at the three JR-STATION sites in northern WSL (IGR, DEM and NOY); the temperature is minimum in January to February

and zero or above zero temperatures are only reached in March. We did not, however, find any consistent pattern between year-to-year spring temperature variation and the occurrence of a spring peak. To determine the cause of the spring peak in emissions, further information is required, such as measurements of atmospheric $\delta^{13}C$-$CH_4$.

**4.2.2 Inter-annual variability**

The inter-annual variability was calculated by first subtracting the mean seasonal cycle (resolved monthly) from the time series for each region, and second, by performing a running average on the residuals with a 6-month time window (see Fig. 12). In North Eurasia, the year-to-year variability in the posterior fluxes is considerable and is much larger than that in

the prior fluxes. In 2005 to 2006, and 2011, the fluxes fell below the $10^{th}$ percentile, while in 2007, 2008, 2010, and 2013, the fluxes reached or exceeded the $90^{th}$ percentile. The inter-annual variability in the WSL is similar to that of all of North Eurasia (R = 0.53) and, in particular, the 2007 anomaly almost entirely originates in the WSL. The year 2007 was particularly warm in the WSL (+1.1°C compared to the 2005 – 2013 annual mean from

ECMWF EI data) and particularly wet (+0.87×$10^{-2}$ $m^3$ $m^{-3}$ soil water volume). Therefore, it is likely that this anomaly is driven by increased $CH_4$ production by wetlands. We find an anomaly for the WSL in 2007 of +3.3 Tg of $CH_4$ (compared to the 2005 – 2013 annual mean), which is similar to that found by Bousquet et al. (2011) of +4 to +5 Tg for the boreal region. This finding further supports the hypothesis that the 2007 anomaly in the atmospheric growth

rate was at least in part due to an increase in boreal wetland emissions as previously suggested (Rigby et al. 2008; Dlugokencky et al. 2009; Bousquet et al. 2011; Bruhwiler et al. 2014).

Although warmer temperature was a factor in the 2007 anomaly in the WSL, there is no significant correlation of temperature with $CH_4$ flux over 2005 – 2013, since positive

temperature anomalies often coincided with negative soil moisture anomalies, which limit $CH_4$ production. In general for WSL, we find a weak correlation of $CH_4$ flux with soil moisture (R = 0.33, p-value = <0.001, see Supplementary Information Fig. 6), and that the correlation increases with a 6-month lag (R = 0.66, p-value = <$1×10^{-12}$) suggesting that wintertime soil moisture may be important for the annual $CH_4$ production.





In North America, again considerable year-to-year variability in the posterior fluxes is seen and exceeds that of the prior. The years 2005 and 2013 had negative anomalies, while 2011 stood out as a strong positive anomaly. The time series for the HBL is strongly correlated with that of North America (R = 0.80), while that of Alberta was moderately correlated (R = 0.59). The year 2011 was a warm year in boreal North America (+0.26°C compared to the 2005 – 2013 annual mean). For the HBL, we find a moderate correlation with soil temperature (R = 0.53, p-value = $<1\times10^{-8}$) but a negative correlation with soil moisture (which arises as temperature and moisture are negatively correlated in the HBL) indicating that soil moisture is not limiting $CH_4$ production. This result contrasts with the result for the WSL, where soil moisture does appear to be a limiting factor. In addition, we looked for correlations between $CH_4$ flux and snow depth and precipitation (from ECMWF EI) but found these to be generally not significant (see Supplementary Information, Fig. 6).

### 4.2.3 Analysis of flux trends

Mann Kendall tests showed significant trends over 2005 – 2013 in northern North America (p-value < 0.01) with a mean rate of increase of 0.38 to 0.57 Tg $y^{-2}$ (range of all scenarios), and, specifically, in the HBL, with mean rate of increase of 0.22 to 0.23 Tg $y^{-2}$ (p-value ≪ 0.001). However, we find no significant trend for Alberta. ECMWF EI data show increasing soil temperature over North America (0.08°C $y^{-1}$) and, especially, the HBL (0.13°C $y^{-1}$), which suggests that the increase in $CH_4$ fluxes is due to wetlands.

Similarly, we find a significant trend (p-value < 0.01) in North Eurasia with mean of 0.76 to 2.50 Tg $y^{-2}$. The upper limit of this range is from S3, which was less well constrained for this region, without S3 the upper limit is 1.09 Tg $y^{-2}$, which we consider more plausible, thus we consider only scenarios S1 and S2 in the following discussion. The North Eurasian trend has approximately equal contributions from northern Europe (EU countries north of 50°N) and Russia of 0.53 to 0.57 Tg $y^{-2}$ and 0.30 to 0.72 Tg $y^{-2}$, respectively. The result for northern Europe contrasts with the prior fluxes, which show a small decrease owing to a reduction in the anthropogenic emissions of -0.07 Tg $y^{-2}$ (according to EDGAR-4.2FT2010). Instead, the increase found in the inversions may be due to wetland sources, a hypothesis that is supported by ECMWF EI data, which show an increase in soil moisture ($0.07\times10^{-2}$ m$^3$ m$^{-3}$ $y^{-1}$). For Russia, on the other hand, the prior fluxes show a substantial increase of 0.30 Tg $y^{-2}$ due to anthropogenic sources and this corresponds to our lower estimate from the inversions. Further





support for an increase in anthropogenic sources is given by British Petroleum energy statistics, which show steady increases in oil and gas production in Russia between 2005 and 2013 of 4% and 12%, respectively, while over the same time period, there is no trend in the ECMWF EI soil moisture or temperature.

In contrast to our study, previous multi-annual inversions have not detected any trend in the high northern latitude $CH_4$ fluxes (Bruhwiler et al. 2014; Bergamaschi et al. 2013). This may be owing to the limited number of observation sites included. As previously mentioned, the inversion of Bergamaschi et al. (2013) included only 7 discrete sampling sites north of 50°N

and the only constraint on fluxes in Northern Eurasia is from continental outflow detected at remote sites, such as Zeppelin on Svalbard. Although the inversion of Bruhwiler et al. (2014) included more sites, as well as in-situ measurements, these were located largely in North America and, again, there were no observations in Siberia. The study of Bloom et al. (2010), conversely, predicted an increase in extra-tropical (45-67°N) and Arctic (>67°N) wetland

fluxes from 2003 to 2007 (the period covered by their study), which was based on a positive correlation of $CH_4$ production and temperature. A recent study by Sweeney et al. (2016) based on atmospheric $CH_4$ measurements at Barrow in Alaska, detected a recent increase (from ~2010) in the late autumn to early winter fluxes from the North Slope of Alaska, which correlates with increasing surface temperatures. However, they did not detect any significant

trend in the mean July to December fluxes. Their study focused on the atmospheric record, and thus did not fully examine climate or ecological changes on the North Slope of Alaska that may help explain the absence of a trend in the July-December fluxes despite the significant increase in surface temperature.

**5. Summary and conclusions**
We have presented spatiotemporally resolved $CH_4$ flux estimates for the high northern latitudes (north of 50°N) from an atmospheric inversion for the period 2005 to 2013. The inversion included observations from in-situ measurement sites in the JR-STATION network in Siberia, the EC network in Canada, as well as the sites, Pallas in Finland, Zeppelin in

Svalbard, Mace Head in Ireland, and ZOTTO in Siberia, and four discrete flask-air sampling sites. This is the first time that these observations have been used together in an atmospheric inversion. We find a $CH_4$ source from the high northern latitudes of 82.0 to 83.6 Tg y$^{-1}$ (the range of S1 and S2) representing ~15% of the global total (i.e., 548 Tg y$^{-1}$ from recent global





inversions (Kirschke et al. 2013)). This is significantly higher than the prior estimates of 64.3 to 67.9 Tg y$^{-1}$ (12% of the global total).

For northern North America, we find an annual mean total source of 16.6 to 17.9 Tg y$^{-1}$,
which is larger than the prior estimates based on EDGAR-4.2FT2010, for anthropogenic emissions, and the land surface models LPX-Bern and LPJ-DGVM, for the wetland fluxes. The regions of the HBL and Alberta were found to be dominating the source. In the HBL, the fluxes are mainly from wetlands and our estimate of 2.7 to 3.4 Tg y$^{-1}$ is close to the prior estimate in scenario S1 (which included wetland flux estimates from LPX-Bern) and to other
inversion estimates, but lower than the prior estimate in S2 (which included wetland flux estimates from LPJ-DGVM). The seasonal cycle in the HBL showed a maximum in August to September, with a rapid decline in fluxes thereafter and near zero fluxes from November to March. In Alberta, our inversions reveal an important source of 5.0 to 5.8 Tg y$^{-1}$, which was found to persist even during winter suggesting that it is of anthropogenic origin, in which case,
current inventories significantly underestimate the emissions.

For North Eurasia, we find an annual mean total source of 52.2 to 55.5 Tg y$^{-1}$. This is significantly larger than the prior estimates, predominantly due to an increase in fluxes in the WSL, from 11.0 to 12.2 Tg y$^{-1}$, a priori, to 19.3 to 19.9 Tg y$^{-1}$, a posteriori. For the WSL, we
estimate an anthropogenic source of $12.7 \pm 3.6$ Tg y$^{-1}$ and a wetland source of $6.9 \pm 3.6$ Tg y$^{-1}$. Anthropogenic emissions in the WSL are dominated by gas and oil production, and our estimate is significantly larger than that of the EDGAR-4.2FT2010 inventory but lower than that of the GAINS model. The seasonal cycle in the WSL has a maximum in July to August, consistent with most land surface models, and shows considerable emissions in winter owing
to anthropogenic sources. For Russia, we found an increasing trend in the fluxes of 0.30 to 0.72 Tg y$^{-2}$ with the lower end of this range corresponding to the trend in the anthropogenic emissions in the EDGAR-4.2FT2010 inventory. The absence of any trend in soil temperature or moisture over our study period, further suggests that the increase is largely due to anthropogenic sources.

Although our study covers only a relatively short period, from 2005 to 2013, notable variations in the year-to-year CH$_4$ fluxes have been identified. In particular, large positive anomalies were seen for the WSL in 2007, and for the HBL in 2011, both due to anomalously high temperatures. Moreover, we detect positive trends in the source from North America and,





specifically, from the HBL, which are correlated with soil temperature. This result may indicate a positive temperature feedback on $CH_4$ emissions in the high latitudes, as expected in the first order based on the temperature dependence of microbial $CH_4$ production. However, on longer timescales, the impact of higher temperature on hydrology and ecosystems in the

boreal and Arctic regions, and thus on $CH_4$ production and oxidation, is very uncertain and an important area of on-going research.

**Datasets**

The observations of atmospheric $CH_4$ mixing ratio used in this paper are available from the

following sources: NOAA ESRL data: http://www.esrl.noaa.gov/gmd/dv/data/, JR-STATION data: http://db.cger.nies.go.jp/portal/geds/atmosphericAndOceanicMonitoring, EC and Teriberka data: http://ds.data.jma.go.jp/gmd/wdcgg/wdcgg.html, Zeppelin data: http://ebas.nilu.no, ZOTTO tower data: on request to J. V. Lavric at MPI-BGC, Pallas station data: on request to T. Aalto at FMI, and AGAGE data: http://agage.mit.edu/data. The

inversion framework, FLEXINVERT, is available from the website: http://flexinvert.nilu.no.

**Competing interests statement**

The authors declare that they have no conflict of interest.

**Acknowledgments**

We thank Colm Sweeney and Ed Dlugokencky for use of the NOAA ESRL data, Peter Bergamaschi for providing TM5 inversion results, and Renato Spahni for providing LPX-Bern wetland fluxes. This study was funded by the Nordic Center of Excellence, eSTICC (eScience Tool for Investigating Climate Change in northern high latitudes), funded by

Nordforsk (grant no. 57001) and the projects, SLICFONIA (emissions of Short-LIved Climate FOrcers Near and In the Arctic), funded by the NORRUSS programme of the Research Council of Norway (grant no. 233642), and EVA (Earth system modeling of climate Variations in the Anthropocene), funded by the Research Council of Norway (grant no. 229771).

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



**Table 1**. Atmospheric measurement sites included in the inversion.

| Station | ID | Network | Latitude | Longitude | Altitude (m) | Time period |
|---|---|---|---|---|---|---|
| Zeppelin* | ZEP | NILU | 78.9°N | 11.9°E | 479 | 2004 – 2013 |
| Tiksi | TIK | NOAA | 71.6°N | 128.9°E | 43 | 2011 – 2013 |
| Teriberka* | TER | MGO | 69.2°N | 35.1°E | 40 | 2004 – 2013 |
| Pallas* | PAL | FMI | 68.0°N | 24.1°E | 565 | 2004 – 2013 |
| Noyabrsk | NOY | JRS | 63.4°N | 75.8°E | 143 | 2005 – 2013 |
| Igrim* | IGR | JRS | 63.2°N | 64.4°E | 72 | 2005 – 2013 |
| Yakutsk | YAK | JRS | 62.1°N | 129.4°E | 287 | 2007 – 2013 |
| ZOTTO | ZOT | MPI | 60.8°N | 89.4°E | 125 | 2009 – 2013 |
| Demyanskoe* | DEM | JRS | 59.8°N | 70.9°E | 138 | 2005 – 2013 |
| Churchill | CHL | EC | 58.8°N | 94.1°W | 35 | 2007 – 2013 |
| Karasevoe* | KRS | JRS | 58.3°N | 82.4°E | 117 | 2004 – 2013 |
| Baltic Sea | BAL | NOAA | 55.5°N | 16.7°E | 28 | 2004 – 2011 |
| Cold Bay* | CBA | NOAA | 55.2°N | 162.7°W | 25 | 2004 – 2013 |
| Lac La Biche* | LLB | EC | 55.0°N | 112.5°W | 546 | 2004 – 2013[a] |
| Azovo | AZV | JRS | 54.7°N | 73.0°E | 150 | 2009 – 2013 |
| Vaganovo | VGN | JRS | 54.5°N | 62.3°E | 285 | 2008 – 2013 |
| East Trout Lake* | ETL | EC | 54.4°N | 105.0°W | 492 | 2005 – 2013 |
| Candle Lake | CDL | EC | 53.9°N | 104.7°W | 489 | 2004 – 2007 |
| Mace Head* | MHD | AGAGE | 53.3°N | 9.9°W | 25 | 2004 – 2013 |
| Fraserdale* | FSD | EC | 49.9°N | 81.6°W | 210 | 2004 – 2013 |
| Chibougamau | CHM | EC | 49.7°N | 74.3°W | 393 | 2007 – 2010 |
| Estevan Point* | ESP | EC | 49.4°N | 126.6°W | 39 | 2004 – 2013[b] |

a. flask samples from NOAA 2004 – 2007 and continuous thereafter
b. flask samples 2004 – 2009 and continuous thereafter
* sites used in scenario S3

**Table 2**. Prior fluxes (units Tg y$^{-1}$) by source type for 2009. For the wetland fluxes, two different models were used to form sets of prior fluxes for scenarios S1 and S2.

| Source Type | | Dataset | | Total | |
|---|---|---|---|---|---|
| | | S1 | S2 | S1 | S2 |
| Natural | Wetlands | LPX-Bern | LPJ-DGVM | 202 | 175 |
| | Termites | Sanderson et al. 1996 | Sanderson et al. 1996 | 19 | 19 |
| | Wild animals | Houweling et al. 1999 | Houweling et al. 1999 | 5 | 5 |
| | Ocean | Lambert et al. 1993 | Lambert & Schmidt 1993 | 17 | 17 |
| | Soil oxidation | LPX-Bern | Ridgwell et al. 1999 | -49 | -38 |
| | Biomass Burn. | GFED3.1 | GFED3.1 | 13 | 13 |
| Anthro-pogenic | Fuel & industry | EDGARv4.2-FT2010 | EDGAR v4.2-FT2010 | 150 | 150 |
| | Enteric ferment. | EDGAR v4.2-FT2010 | EDGAR v4.2-FT2010 | 101 | 101 |
| | Waste | EDGAR v4.2-FT2010 | EDGAR v4.2-FT2010 | 61 | 61 |
| | Rice cultivation | LPX-Bern | EDGAR v4.2-FT2010 | 36 | 38 |
| Total | | | | 556 | 541 |





**Table 3**. Inversion cost a priori and a posteriori shown for 2009.

| Scenario | Cost a priori | Cost a posteriori |
|---|---|---|
| S1 | 57737 | 17531 |
| S2 | 57278 | 17378 |
| S3 | 51472 | 13352 |

**Table 4**. Mean prior and posterior flux totals (units Tg y$^{-1}$) for each inversion scenario and comparison to independent inversion estimates for 2005 to 2010 (the overlapping period).

| Inversion | No. in situ + (flask) sites | North America | | North Eurasia | |
|---|---|---|---|---|---|
| | | Prior | Posterior | Prior | Posterior |
| S1 | 17 + (5) | 9.5 ± 5.1 | 16.6 ± 1.1 | 44.4 ± 12.5 | 55.2 ± 2.5 |
| S2 | 17 + (5) | 14.2 ± 5.1 | 17.9 ± 1.1 | 43.3 ± 12.5 | 52.2 ± 2.5 |
| S3 | 10 + (2) | 9.5 ± 5.1 | 17.1 ± 1.3 | 44.4 ± 12.5 | 59.5 ± 3.2 |
| MACC NOAA[a] | (7) | - | 14.0 | - | 34.0 |
| CT-CH$_4$[b] | 6 + (10) | 7.5 | 8.1 | 60.3 | 49.7 |

a. Bergamaschi et al. (2013)
b. Bruhwiler et al. (2014)



**Figures**

**Figure 1**. Map showing the atmospheric measurement sites used in the inversion. The grey dashed line indicates the southern boundary of the inversion domain at 50°N. Flask-air sampling sites are indicated in blue and in situ sites in red.

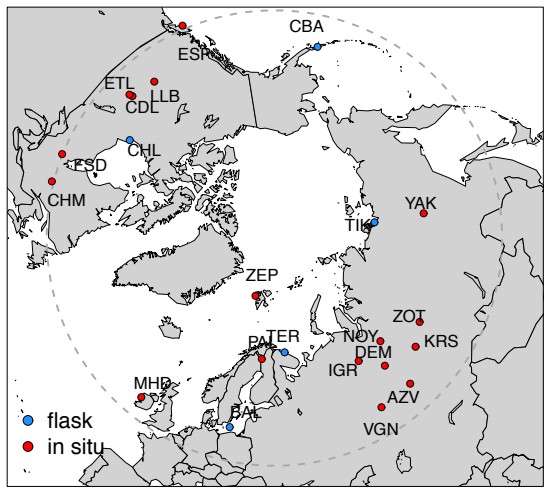



**Figure 2**. The annual mean emission sensitivity of all sites shown for 2009 (units of $\log_{10}(s\ m^3\ kg^{-1})$) with the southern domain boundary shown by the grey dashed line (a) and the grid used in the inversion based on the mean emission sensitivity (b). The sites used in the inversion are indicated by the black points.

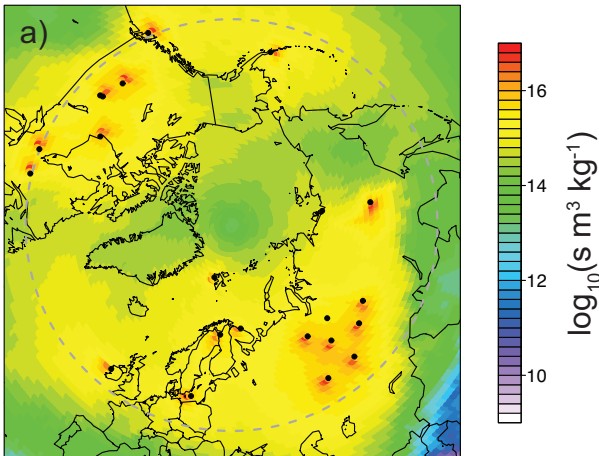

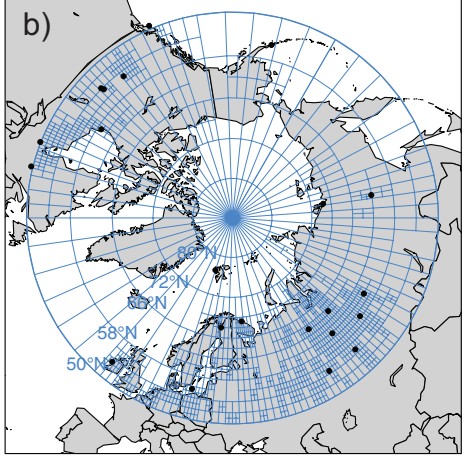





**Figure 3**. Example of the initial concentration field (units ppb) for January 2009 interpolated from observations in the NOAA network (sites indicated by the black points) (a) and an example shown for site IGR (black dot) of the sensitivity to the initial concentrations (percentage) after 10 days backward calculation (mean January-2009).

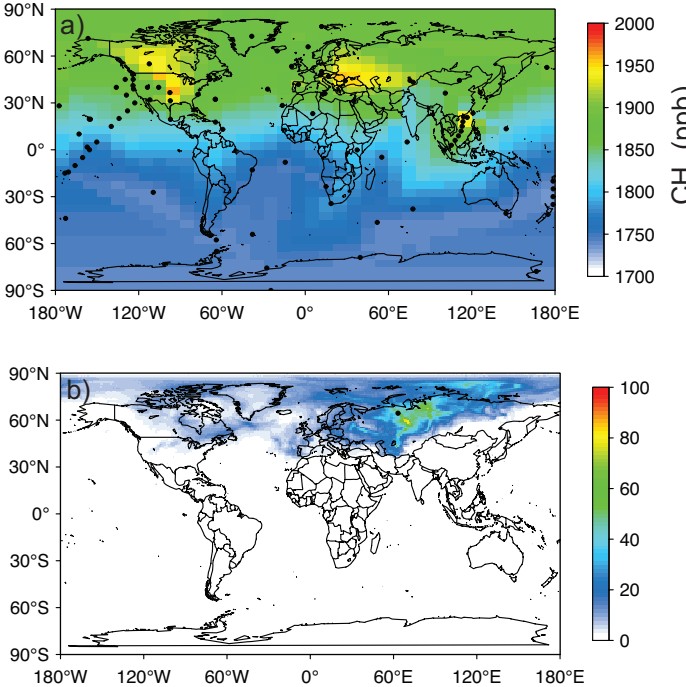



**Figure 4**. Example of measurements of CH₄, temperature at two heights on the tower, and wind speed, at IGR in January 2009 (a) and July 2009 (b). Also shown in the upper plots, are CH₄ concentrations simulated with prior fluxes using FLEXPART driven with ECMWF EI (blue) and NCEP FNL (red) meteorological analyses and a sensitivity test using ECMWF EI but with no minimum PBL height set in FLEXPART (green) (note that this is on top of the blue line). The vertical shading indicates periods when data were filtered out.

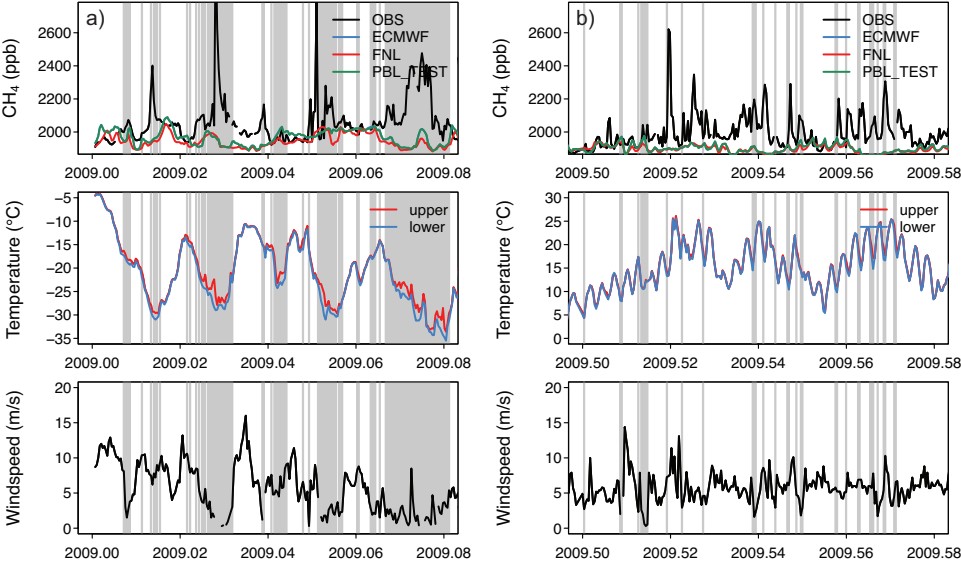



**Figure 5**. Taylor diagrams for the comparison of the prior (open circles) and posterior (solid circles) simulated concentrations with the observations for 2009 (the radius indicates the normalized standard deviation and the angle the correlation coefficient). The results for the two scenarios are shown (i.e., S1 in magenta and S2 in blue).

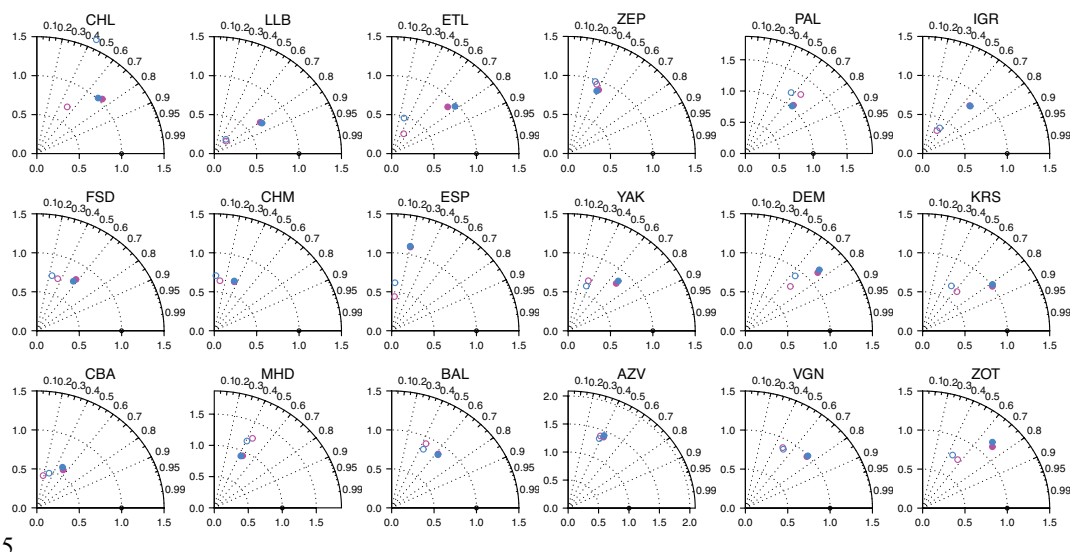




**Figure 6**. Comparison of simulated concentrations from scenario S1 with independent observations from NOAA aircraft campaigns at Poker Flats (PFA), Estevan Point (ESP) and East Trout Lake (ETL) in Canada. The three rows of dots from top to bottom are the comparison for the mean of data between 4 to 10 km, 1 to 4 km and 0 to 1 km. Prior concentrations (a) and posterior concentrations (b).

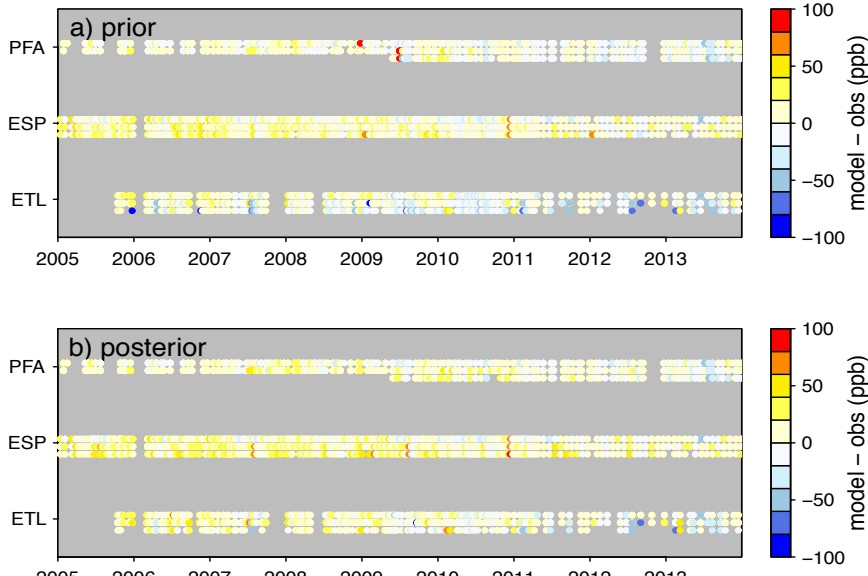





Figure 7. Annual mean posterior fluxes of $CH_4$ shown for 2009 for scenario S1 (a), the difference between the posterior and prior fluxes for S1 (b), the posterior fluxes for scenario S2 (c) and the difference for S2 (d), the posterior fluxes for scenario S3 (e) and the difference for S3 (f). The bordered areas are Alberta in western Canada, the Hudson Bay Lowlands (HBL) in eastern Canada, and the Western Siberian Lowlands (WSL). (Units are $gCH_4$ m$^{-2}$ day$^{-1}$).

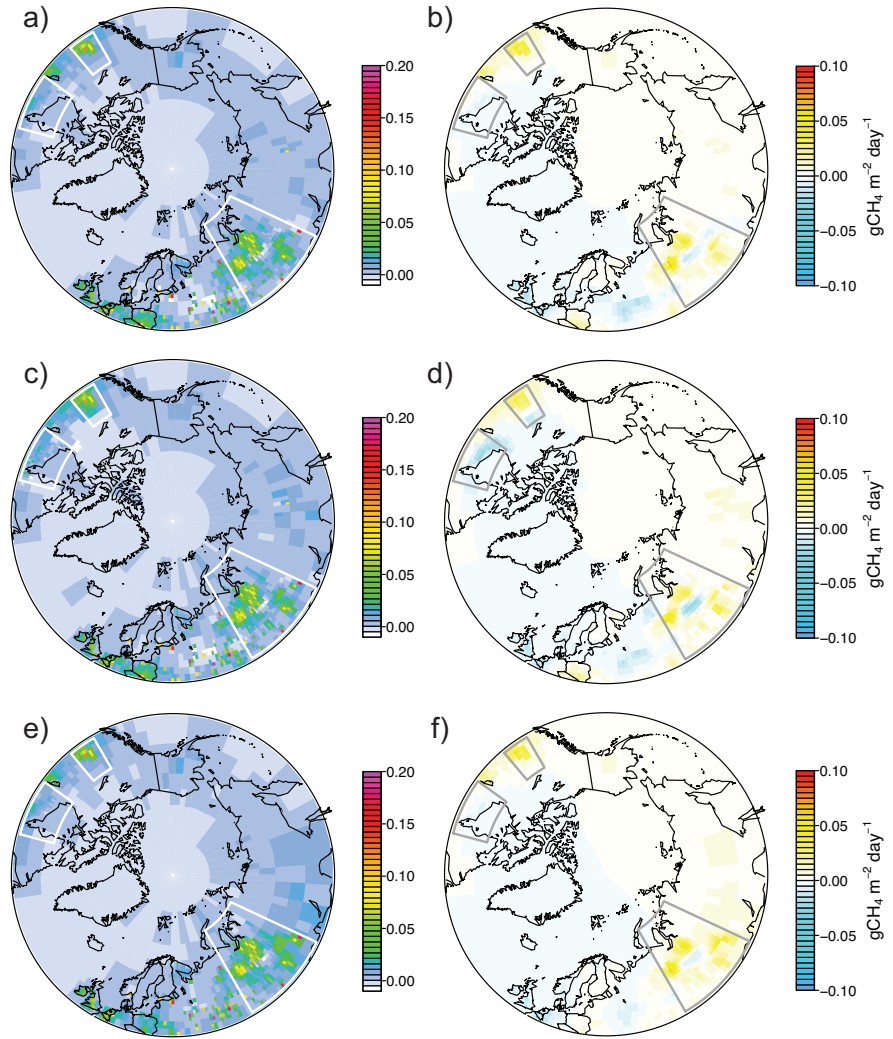





**Figure 8**. Fractional uncertainty reduction for 2009 for scenarios S1 and S2 (a), and S3 (b).

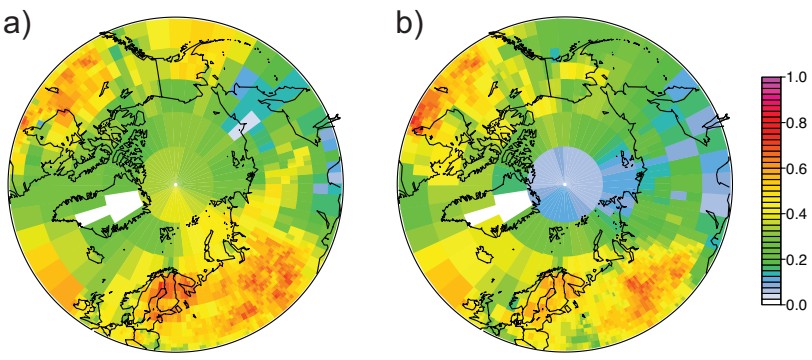





**Figure 9**. Area integrated CH$_4$ fluxes (units TgCH$_4$ y$^{-1}$) from the three scenarios shown monthly for northern North America and North Eurasia. The prior fluxes are shown by the dashed lines and the posterior fluxes by the solid lines (note that the prior fluxes used in scenarios S1 and S3 are the same). The grey shading indicates the prior uncertainty (shown only for the S1 prior) and the coloured shading indicates the posterior uncertainty.

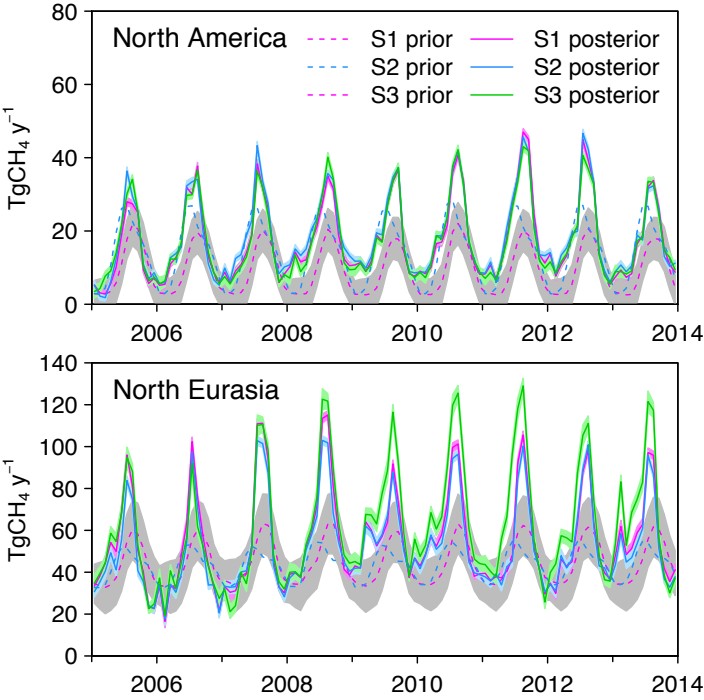

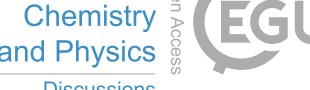



**Figure 10**. Mean posterior fluxes of $CH_4$ (top row) and posterior – prior differences (bottom row) shown for scenario S1 for winter 2008-2009 (DJF), spring 2009 (MAM), summer 2009 (JJA), and autumn 2009 (SON). (Units are $gCH_4\ m^{-2}\ day^{-1}$).

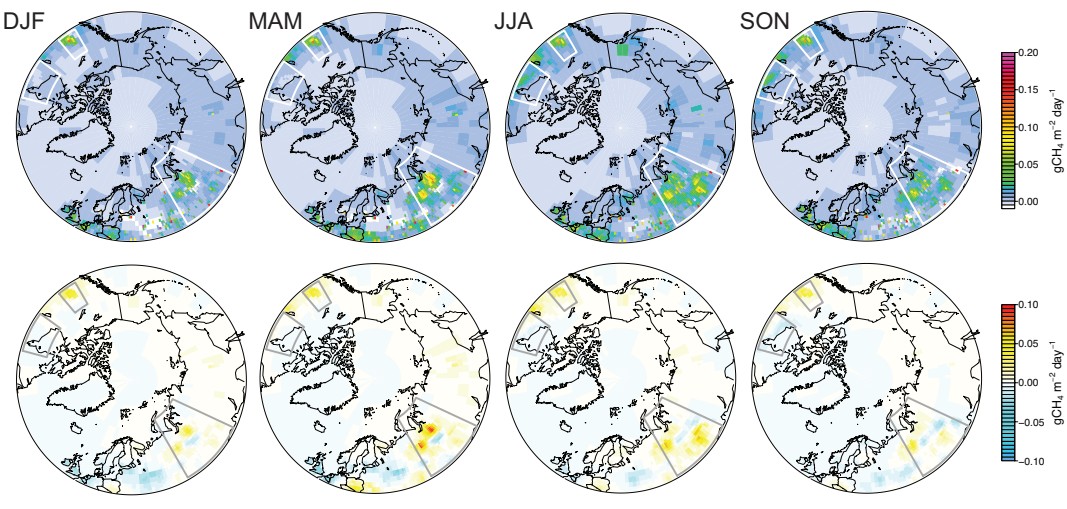

**Figure 11.** Mean seasonal cycles (2005 – 2013) of the integrated CH$_4$ flux for different regions (units Tg y$^{-1}$). The grey shading indicates the standard deviation of the monthly fluxes. The solid lines show the posterior fluxes and the dashed lines show the prior fluxes. Magenta is for scenario S1 (including wetland fluxes from LPX-Bern) and blue is for S2 (including wetland fluxes from LPJ-DGVM) and black (dashed) is the prior anthropogenic fluxes.

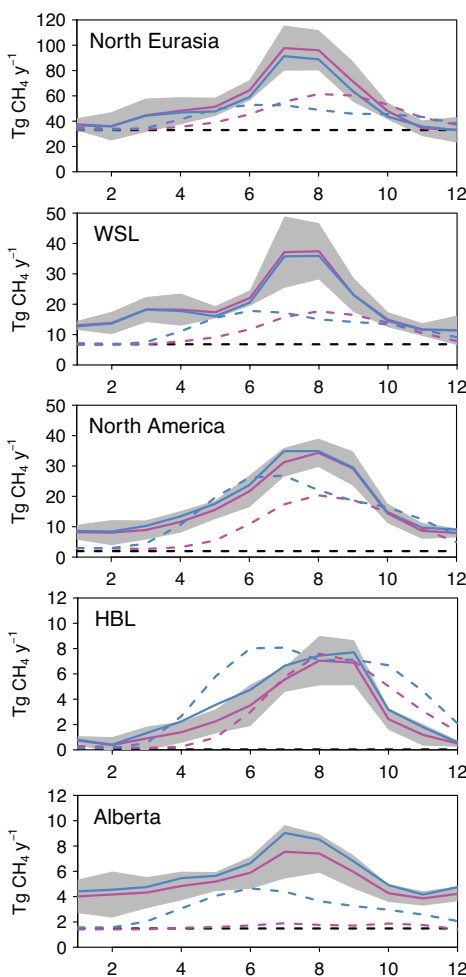


**Figure 12**. Inter-annual variability in CH$_4$ fluxes (units Tg y$^{-1}$) for North Eurasia, WSL, northern North America, HBL and Alberta. The inter-annual variability was calculated by subtracting the mean seasonal cycle (resolved monthly) from the time series for each region and performing a running average on the residuals with a 6-month time window. The solid lines are the posterior fluxes from S1 (magenta), S2 (blue) and S3 (green) and the dashed lines are the prior fluxes from S1 and S2. The shading indicates the posterior uncertainty.

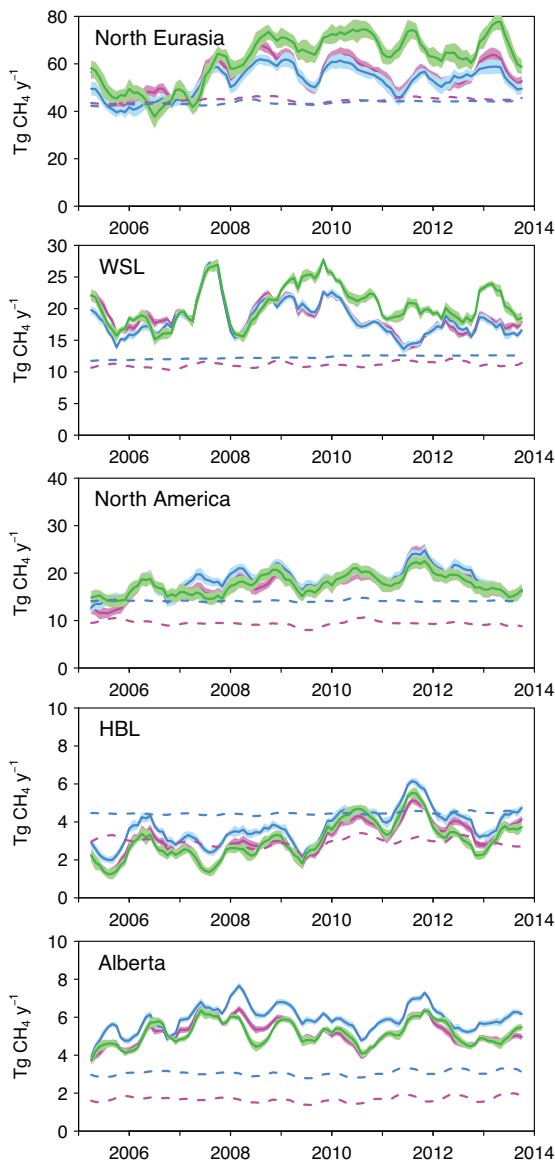