# Peer review of "Methane fluxes in the high northern latitudes for 2005 – 2013 estimated using a Bayesian atmospheric inversion"

_Atmospheric Chemistry and Physics, 2016_

## Referee Comment (RC1) · Anonymous Referee #1 · 26 Oct 2016

General comments

The manuscript by Thompson et al. presents estimates for CH4 emission fluxes in northern high latitudes for the time period 2005 to 2013 using an atmospheric inversion. The study is based on the Lagrangian transport model FLEXPART and measurement data from 22 observational sites in northern high latitudes. The sensitivity of the results to prior estimates of wetland emission fluxes and to the number of measurement sites included in the inversion is also investigated.

The atmospheric methane budget is an important topic and under ongoing scientific debate. There are large uncertainties on the total amount of emissions, but also on the importance of individual source categories, as well as on their change over time.

[Figure]

Especially natural emission estimates based on process-oriented models show a large uncertainty range. Atmospheric inversions are a widely used technique to gain new insights into methane fluxes. The present study makes use of a relatively new observational network in northern high latitudes. Some of these data have already been used for regional inversions, but not in a combined study for the total area north of 50°N. Therefore, the study makes an important contribution to an improved picture of CH4 emission fluxes and provides further insights in the quality of available CH4 emission inventories. The results could also be used for evaluation and improvement of wetland emission models.

The manuscript is generally well written, the figures are well prepared and the results are discussed in an appropriate way. My main questions are related to the different time scales applied in the inversion, e.g. monthly or even annual emission fluxes, monthly initial methane mixing ratios, but 10-day backward trajectories and in-situ measurements. I have a couple of remarks and questions on the applied method as well as some suggestions for improvements (see below). After taking these comments into account I recommend the paper for publication.

Specific comments

- Sect. 2.3: I would like to see Fig. 5 of the supplement in the main paper. Since the wetland data set is the main difference between S1 and S2, it is interesting to see how their spatial distributions differ.

- P8, l23-25: How does the dry soil uptake provided by Ridgwell et al. compare to the LPX-Bern model in terms of absolute numbers and seasonality? Is there a strong interannual variability in the uptake calculated by LPX-Bern, which is neglected by the climatology?

- Which meteorological input data was used for the two land surface models? Also ERA-Interim?

- P9, l8/9: Is there any reason for using 50% of the prior flux as uncertainty? How sensitive are your results to that assumption? And how often do your minimum and maximum thresholds apply?

- P9, l20: What is the reason for calculating backward trajectories for 10 days? The prior emission fluxes are provided on a monthly or even annual basis. Are these 10 days an average transport or mixing time scale? Or is that an attempt to minimize the impact of chemical methane loss on the results, i.e. to minimize the uncertainties from using a pre-calculated OH field?

- P9, l30-32: The calculation of methane loss by the reaction with OH is based on pre-calculated OH fields from the GEOS-Chem model. How does the GEOS-Chem OH field compare to other models? Could you provide a reference?

- P10, l24: Here you state that the initial mixing ratios are calculated at a monthly temporal resolution. Again I have some difficulties to bring the different time resolutions together: monthly mean prior fluxes and initial fields, 10-days backward trajectories released every 3 hours, results compared to in-situ measurements. This approach certainly reduces uncertainties resulting from shortcomings in the representation of the chemical sink, the dry soil uptake, mixing processes, etc., which would become more important over a longer simulation period. However, it also neglects short-term variations that are visible in the observational data, e.g. in Fig. 4. Does that have an impact on your inversion results or not, because you are looking at total emissions over a month? My question probably reflects my limited knowledge of the FLEXINVERT framework.

- P13, l5-7: I do not understand this sentence. Please re-formulate.

- P13, l14/15: Does the low bias in the prior mixing ratios indicate any flaws in the method used for calculating the initial and background mixing ratios?

- P14, l4-5: On page 13 you state that the comparison of the inversion results with

independent observations is a better indicator for the performance of the inversion than the Taylor diagrams shown in Fig. 5. Figure 6 shows at two out of three independent sites only a modest improvement, which by the way is hard to see from Figure 6 (you might want to change the color scale). What does that mean for the quality of the inversion? Any explanation for that? Please comment on this.

- P14, l29: How is the uncertainty in each grid cell defined? Is $\sigma$prior calculated as defined on page 9, line 9-10?

- P15, l1/2: Why is the largest uncertainty reduction found in Europe, western Siberia and Canada? Please comment on this.

- P15, discussion of Fig. 9: The posterior fluxes in Fig. 9 show several secondary maxima in the annual cycle. On page 19, l22/23 you mention a small secondary peak in March. Is this the same feature as seen in Fig. 9? It would be great if you could comment on these peaks or at least refer to the later discussion.

- Sect. 4.1: The discussion here is rather lengthy and it is hard to keep an overview over the various studies and flux estimates. I would prefer to see a table or a figure, e.g. a bar chart, summarizing the various inversion results.

- Sect. 4.2: I would suggest to merge this section with Sect. 4.1 to make the discussion a bit shorter and therefore clearer.

- P21, l31-32: Are the wetland models LPX-Bern and LPJ-DGVM also driven by ECMWF EI data? I remember that some of these models are driven by CRU data. In that case it might be misleading to explain the increase in the wetland source by an increase in soil moisture found in EI data.

- Table 1: I would like to see the numbers for the region of >50°N as well.

- Table 3: There is only one reference to Table 3 in the text, discussing the lower cost of the prior flux estimate in S2 compared to the other priors. The other values given in the table are not discussed. Is this table really necessary? And what is actually listed

in Table 3? How is the cost defined? Does it come with any units?

- Fig. 7 and Fig. 10: I think the color scale could be improved, especially for the difference plots. It is hard to distinguish the different bluish shades.

---

## Referee Comment (RC2) · Anonymous Referee #2 · 8 Dec 2016

This manuscript is a well-written, clearly presented description of an inverse modelling study focussing on high latitude sites. Other studies have focussed on this region, but this study is unique for the collection of sites used in the inversion, as well as the significant time period covered in the inversion. It also has somewhat higher spatial resolution than what is commonly used for global inversion studies, making use of different grid resolutions to reflect the measurements constraint on the fluxes for different regions.

The combination of the extensive collection of measurements employed, the still uncertain and hotly contested topic of the (Arctic) methane budget, and the inversion approach make this study appropriate in terms of content for ACP. The results are

interesting, and can be used to evaluate the performance of wetland models, and point to potential shortcomings of the anthropogenic emission inventories. Overall the manuscript is very well presented, although the discussion does run a bit long. (I support the previous reviewer's suggestion to summarize the key results in a table if possible, and condense the text somewhat.) I have only a few relatively minor suggestions, as outlined below.

Why was GFED3 at 0.5 degree/monthly resolution used? There are certainly newer versions of GFED, and temporal resolutions of higher than monthly are the norm now. (The emissions from GFED are still given at monthly resolution, but there are daily and even 3-hourly fields to scale the monthly emissions appropriately.) When trying to capture the synoptic scale variability in methane fluxes, it really does make a difference if the fire burned on July 1-5 vs. July 20-25, which at this point you're neglecting. Because the fire flux is of a relatively small term in your budget the impact is likely not critical, but it is an easily rectifiable methodological shortcoming. If not for this study, than certainly for future work.

In Figure 2 (and in the model in general), is this sensitivity shown only for the lowest model level? Or is the addition from fluxes from other grid boxes within the PBL included?

Figure 5 rather underwhelms in terms of improvements through inversion, however Figure 6 shows (for one set of independent data) a clear improvement in the bias from prior to posterior. Would it be possible to include the mean or median bias per site in Figure 5? This can be done with, for instance, the size of the marker. It would be interesting to see if this was the case for other stations as well, which would help support the argument that the posterior is a significant improvement on the prior, and the flux corrections are really robust.

P11, L35: I think the section starting with "For other continental sites..." should go immediately after the previous paragraph i.e. the comment about the IGR representation error should come later. Or at least remind the reader what "this criterion" is, as it doesn't follow clearly as it's written now.

The citation to (Winderlich 2010) is incorrect. There are several full references missing the full stop.

---

## Author Comment (AC1) · 10 Jan 2017

**Response to Review Comments**

**Reviewer 1**

We thank the reviewer for his/her thoughtful and constructive comments and we reply to each of these below.

**General comments**

The manuscript by Thompson et al. presents estimates for CH4 emission fluxes in northern high latitudes for the time period 2005 to 2013 using an atmospheric inversion. The study is based on the Lagrangian transport model FLEXPART and measurement data from 22 observational sites in northern high latitudes. The sensitivity of the results to prior estimates of wetland emission fluxes and to the number of measurement sites included in the inversion is also investigated.

The atmospheric methane budget is an important topic and under ongoing scientific debate. There are large uncertainties on the total amount of emissions, but also on the importance of individual source categories, as well as on their change over time.

Especially natural emission estimates based on process-oriented models show a large uncertainty range. Atmospheric inversions are a widely used technique to gain new insights into methane fluxes. The present study makes use of a relatively new observational network in northern high latitudes. Some of these data have already been used for regional inversions, but not in a combined study for the total area north of 50N. Therefore, the study makes an important contribution to an improved picture of CH4 emission fluxes and provides further insights in the quality of available CH4 emission inventories. The results could also be used for evaluation and improvement of wetland emission models.

The manuscript is generally well written, the figures are well prepared and the results are discussed in an appropriate way. My main questions are related to the different time scales applied in the inversion, e.g. monthly or even annual emission fluxes, monthly initial methane mixing ratios, but 10-day backward trajectories and in-situ measurements. I have a couple of remarks and questions on the applied method as well as some suggestions for improvements (see below). After taking these comments into account I recommend the paper for publication.

**Specific comments**

- Sect. 2.3: I would like to see Fig. 5 of the supplement in the main paper. Since the wetland data set is the main difference between S1 and S2, it is interesting to see how their spatial distributions differ.

We have moved Fig. 5 from the supplement to Fig. 3 of the manuscript.

- P8, 123-25: How does the dry soil uptake provided by Ridgwell et al. compare to the LPX-Bern model in terms of absolute numbers and seasonality? Is there a strong interannual variability in the uptake calculated by LPX-Bern, which is neglected by the climatology?

Globally soil uptake in LPX-Bern is larger (49 Tg/y) than in Ridgwell et al. (38 Tg/y) and in the high latitudes  $>50^{\circ}N$  (6.5 and 4.3 Tg/y, respectively, see also Table 2). The seasonal

cycles have a similar shape but LPX-Bern has more uptake in the summer (see figure). Interannual variability in the LPX-Bern soil uptake is small for the high northern latitudes; variations in the annual mean are less than 4%, therefore using a climatology should have little impact.

LPX-Bern used CRU climate date (version TS 3.23) and LPJ-DGVM used monthly mean temperature, cloud fraction and total precipitation (see Bergamaschi et al., 2007)

- P9, 18/9: Is there any reason for using 50% of the prior flux as uncertainty? How sensitive are your results to that assumption? And how often do your minimum and maximum thresholds apply?

We chose to use 50% for the uncertainty weighting on each grid cell, but the total uncertainty for the domain was scaled to 15 Tg/y (~25%) as plus/minus this value covers the approximate range of inventory and land-surface model estimates (see section 2.3). The sensitivity of the inversion to the prior flux uncertainty depends on the how strong the atmospheric constraint is. In this study, we found the inversion to be not very sensitive to the prior uncertainty estimate. Note, that the prior uncertainty range is not a limit or threshold, but specifies the pdf of the prior fluxes. Figure 7 in the manuscript shows how the posterior fluxes differ from the prior ones. For example, for the region North America (i.e. Canada + Alaska) the posterior fluxes are outside the range given by prior S1 but within the range given by the prior S2 (see Table 4).

- P9, 120: What is the reason for calculating backward trajectories for 10 days? The prior emission fluxes are provided on a monthly or even annual basis. Are these 10 days an average transport or mixing time scale? Or is that an attempt to minimize the impact of chemical methane loss on the results, i.e. to minimize the uncertainties from using a pre-calculated OH field?

Ten days was considered an appropriate time scale for the backward trajectories because the longer backwards in time virtual particles (or trajectories) are followed the further they travel

from the point of observation, and the influence of fluxes further from the point of observation have less influence on that observation. This can also be expressed in terms of the degree of dispersion of the virtual particles the further they are traced backwards in time. After several days the particles are generally well dispersed (mixed) and the influence of fluxes from a specific location becomes very small.

Note also that the 10-days are not an average. We calculate trajectories every 3-hours and each trajectory is followed for 10-days. In summary, the surface influence (or emission sensitivity) is calculated every 3-hours.

- P9, 130-32: The calculation of methane loss by the reaction with OH is based on precalculated OH fields from the GEOS-Chem model. How does the GEOS-Chem OH field compare to other models? Could you provide a reference?

Over 10-days the loss due to oxidation by OH is very small (~1 ppb) so much less than the overall uncertainty in the observation space. We do not have access to other OH fields, so we cannot directly compare GEOS-Chem to other models, however, the influence of the OH field will be very small since the loss is over 10 days is very small. The reference for the GEOS-Chem OH fields is: Bey I, Jacob DJ, Yantosca RM, et al.: J. Geophys. Res., 106, 19, 23073-23095, 2001. We have added this reference to the manuscript.

- P10, l24: Here you state that the initial mixing ratios are calculated at a monthly temporal resolution. Again I have some difficulties to bring the different time resolutions together: monthly mean prior fluxes and initial fields, 10-days backward trajectories released every 3 hours, results compared to in-situ measurements. This approach certainly reduces uncertainties resulting from shortcomings in the representation of the chemical sink, the dry soil uptake, mixing processes, etc., which would become more important over a longer simulation period. However, it also neglects short-term variations that are visible in the observational data, e.g. in Fig. 4. Does that have an impact on your inversion results or not, because you are looking at total emissions over a month? My question probably reflects my limited knowledge of the FLEXINVERT framework.

To answer this question we explain briefly how our inversion method works but for a detailed description please see sections 2.1, 2.4 and 2.5 of the manuscript (and also Thompson and Stohl, GMD, 2014).

Our transport model, FLEXPART, represents the atmospheric transport at 3-hourly intervals and reproduces the short-term variations in the observational data very well (see Fig. 1 of the supplement). Please note that we do account for atmospheric mixing, the soil uptake of CH4, as well as the CH4 loss from oxidation by OH. With the prior fluxes, we achieve  $R^2 > 0.3$  for the comparison of the modeled and observed mixing ratios using the daily afternoon averages (this is at all sites except CHM and ESP which have very low variability). All observations in a given month (on average about 300 observations) are used to optimize the fluxes in that month. This gives the average monthly fluxes a posteriori (the prior and posterior flux resolution is the same). We assume that the fluxes are not changing substantially over the course of one month, and at the spatial scale of the fluxes (1×1 degree) this is a reasonable assumption. In summary, we do not neglect short-term variations in the observations at all.

3

- P13, 15-7: I do not understand this sentence. Please re-formulate.

We have reformulated this sentence.

- P13, 114/15: Does the low bias in the prior mixing ratios indicate any flaws in the method used for calculating the initial and background mixing ratios?

Please note that we discuss here normalized standard deviation (NSD) and not bias. The NSD is generally less than one, indicating that the prior modeled mixing ratios have less variability than the observations. Since the variability is largely due to the influence of fluxes within the domain (and considering that the prior model has a reasonably good correlation with the observations of  $R^2 > 0.3$ ), the low NSD cannot be explained by errors in the background mixing ratios.

- P14, 14-5: On page 13 you state that the comparison of the inversion results with independent observations is a better indicator for the performance of the inversion than the Taylor diagrams shown in Fig. 5. Figure 6 shows at two out of three independent sites only a modest improvement, which by the way is hard to see from Figure 6 (you might want to change the color scale). What does that mean for the quality of the inversion? Any explanation for that? Please comment on this.

Of the three independent sites (i.e. not included in the inversion) we see some improvement at all sites but especially at ETL where the correlation improved from  $R^2 = 0.33$  to 0.37. The other two sites, PFA and ESP, are less sensitive to the fluxes in the domain (ESP is a coastal site and PFA is not in an area where there are significant fluxes), hence we do not expect to see a big change in the modeled mixing ratios at these sites with posterior fluxes. Unfortunately, there are not very many observations in the high latitudes, so we chose to include almost all data in the inversion, leaving only these sites for the validation. We also looked at HIPPO aircraft data but found that these were not sensitive enough to the fluxes to be useful for the validation (HIPPO measurements were made over the northern Pacific Ocean and in the mid to upper troposphere).

- P14, 129: How is the uncertainty in each grid cell defined? Is prior calculated as defined on page 9, line 9-10?

The calculation of the prior uncertainty for each grid-cell is described on page 9 lines 8-14. The posterior uncertainty for each grid-cell is calculated as the 1-sigma standard deviation of all inversion results in a Monte Carlo ensemble (see page 5 lines 25-26).

- P15, 11/2: Why is the largest uncertainty reduction found in Europe, western Siberia and Canada? Please comment on this.

This is because these are the regions with the most observations, i.e., with the strongest constraint (see Fig. 1 and Fig. 2a).

- P15, discussion of Fig. 9: The posterior fluxes in Fig. 9 show several secondary maxima in the annual cycle. On page 19, 122/23 you mention a small secondary peak in March. Is this the same feature as seen in Fig. 9? It would be great if you could comment on these peaks or at least refer to the later discussion.

Yes, the secondary maxima in spring (Fig. 9) are the same feature seen for North Eurasia (and more distinctly for WSL) in Fig. 11. In Fig. 9, there also appears to be a secondary spring

maximum in North America for some years (e.g. 2008 and 2009) but this does not appear in the mean seasonal cycle for North America. We have now added a reference on page 15 to the discussion on page 19.

- Sect. 4.1: The discussion here is rather lengthy and it is hard to keep an overview over the various studies and flux estimates. I would prefer to see a table or a figure, e.g. a bar chart, summarizing the various inversion results.

We have included a new table (in the revision this is Table 4) to summarize our main results, and we have shortened the discussion somewhat.

- Sect. 4.2: I would suggest to merge this section with Sect. 4.1 to make the discussion a bit shorter and therefore clearer.

We have considered your suggestion to combine these two sections, but we rather prefer to keep these separated as we think it is easier for the reader to follow when the discussion is separated under different sub-headings. We have in any case shortened the discussion.

- P21, l31-32: Are the wetland models LPX-Bern and LPJ-DGVM also driven by ECMWF EI data? I remember that some of these models are driven by CRU data. In that case it might be misleading to explain the increase in the wetland source by an increase in soil moisture found in EI data.

LPX-Bern is driven by CRU climate data and LPJ-DGVM uses monthly mean meteorology and is a climatology. Regardless of which prior flux estimate we use, the inversions indicate that the fluxes are increasing over the period 2005 – 2013. This increase is obviously not seen in LPJ-DGVM (it is a climatology) and it is also not seen in LPX-Bern. On page 21, we hypothesize that the trend in northern North America and the HBL, as found by the inversion, could be due to an increase in soil temperature, while the increase in northern Europe could be due to an increase in soil moisture. ECMWF EI data show an increase in soil temperature in northern North America and an increase in soil moisture in northern Europe.

- *Table 1: I would like to see the numbers for the region of* >50\_N *as well.*

We have added the numbers for the region  $>50^{\circ}$ N to Table 2 (note: we think the reviewer rather refers to Table 2).

- Table 3: There is only one reference to Table 3 in the text, discussing the lower cost of the prior flux estimate in S2 compared to the other priors. The other values given in the table are not discussed. Is this table really necessary? And what is actually listed in Table 3? How is the cost defined? Does it come with any units?

We agree that Table 3 is not necessary in the manuscript and we have now moved it to the supplement. The cost is the value of the cost function (see Eq. 1) and quantifies the mismatch between modeled and observed mixing ratios as well as the prior and posterior fluxes. The cost is unitless.

- Fig. 7 and Fig. 10: I think the color scale could be improved, especially for the difference plots. It is hard to distinguish the different bluish shades.

We have changed the colour scale for the plots of the differences between posterior and prior fluxes (Fig. 7 and Fig. 10) as we agree that in these figures the variations in the blue colours were not so clear.

**Review 2**

We thank the reviewer for his/her thoughtful and constructive comments and we reply to each of these below.

**General comments**

This manuscript is a well-written, clearly presented description of an inverse modeling study focussing on high latitude sites. Other studies have focussed on this region, but this study is unique for the collection of sites used in the inversion, as well as the significant time period covered in the inversion. It also has somewhat higher spatial resolution than what is commonly used for global inversion studies, making use of different grid resolutions to reflect the measurements constraint on the fluxes for different regions. The combination of the extensive collection of measurements employed, the still uncertain and hotly contested topic of the (Arctic) methane budget, and the inversion approach make this study appropriate in terms of content for ACP. The results are interesting, and can be used to evaluate the performance of wetland models, and point to potential shortcomings of the anthropogenic emission inventories. Overall the manuscript is very well presented, although the discussion does run a bit long. (I support the previous reviewer's suggestion to summarize the key results in a table if possible, and condense the text somewhat.) I have only a few relatively minor suggestions, as outlined below.

Following the first reviewer's suggestion, we have added a table summarizing the main results (Table 4 in the revised manuscript). We have also shortened the discussion section.

Why was GFED3 at 0.5 degree/monthly resolution used? There are certainly newer versions of GFED, and temporal resolutions of higher than monthly are the norm now. (The emissions from GFED are still given at monthly resolution, but there are daily and even 3-hourly fields to scale the monthly emissions appropriately.) When trying to capture the synoptic scale variability in methane fluxes, it really does make a difference if the fire burned on July 1-5 vs. July 20-25, which at this point you're neglecting. Because the fire flux is of a relatively small term in your budget the impact is likely not critical, but it is an easily rectifiable methodological shortcoming. If not for this study, than certainly for future work.

When we commenced this study GFED3 was the most recent dataset available for fire emissions and GFED4 and GFED4s were released later. As the reviewer mentions, fires are a relatively small source of CH4 for the Arctic region, therefore, using a more recent dataset in the prior emissions estimate would not make any notable difference to the results, especially considering that we find little sensitivity of the posterior fluxes to the prior estimate, as shown by the comparison of scenarios S1 and S2. As for using daily, or higher temporal resolution, estimates, if the timing of the emissions is correct in the prior estimate, then a better a priori fit to the observations may be achieved, however, if the timing is wrong, then the fit may actually deteriorate compared to the lower temporal resolution prior estimate.

In Figure 2 (and in the model in general), is this sensitivity shown only for the lowest model

level? Or is the addition from fluxes from other grid boxes within the PBL included?

The emission sensitivity is shown for the lowest level, i.e., 0 - 400 m. All fluxes are assumed to occur in this layer.

Figure 5 rather underwhelms in terms of improvements through inversion, however Figure 6 shows (for one set of independent data) a clear improvement in the bias from prior to posterior. Would it be possible to include the mean or median bias per site in Figure 5? This can be done with, for instance, the size of the marker. It would be interesting to see if this was the case for other stations as well, which would help support the argument that the posterior is a significant improvement on the prior, and the flux corrections are really robust.

We have added the mean error in a table in the Supplementary Information (Table 2). We decided not to include this information in Figure 5 (in the revised manuscript this is Figure 6) as we thought that this would not be clearly visible from the size of the points.

P11, L35: I think the section starting with "For other continental sites..." should go immediately after the previous paragraph i.e. the comment about the IGR representation error should come later. Or at least remind the reader what "this criterion" is, as it doesn't follow clearly as it's written now.

We have moved the section starting with "For other continental sites" to directly below the previous paragraph and the sentence about IGR to the end of the paragraph.

7

The citation to (Winderlich 2010) is incorrect.

We have corrected this citation.

There are several full references missing the full stop.

We have added the full stops

**Methane fluxes in the high northern latitudes for 2005 – 2013 estimated using a Bayesian atmospheric inversion**

Rona L. Thompson1, Motoki Sasakawa2, Toshinobu Machida2, Tuula Aalto3, Doug Worthy4, Jost V. Lavric5,6, Cathrine Lund Myhre1 and Andreas Stohl1

1NILU - Norwegian Institute for Air Research, Kjeller, Norway 2National Institute for Environmental Studies, Tsukuba, Japan 3Finnish Meteorological Institute (FMI), Helsinki, Finland

4Environment Canada, Toronto, Canada 5Max Planck Institute for Biogeochemistry, Jena, Germany 6Integrated 
[revised manuscript text omitted]

20

25

Large fluxes of CH4 from the ocean to the atmosphere have been reported for the East Siberian Arctic Shelf (ESAS) with a source estimated to total 17 Tg y-1 representing  $\sim$ 3% of the global source to the atmosphere (Shakhova et al. 2010; Shakhova et al. 2015). Although our inversion has only a modest reduction in uncertainty in the ESAS region (see Fig. 9) we do not find any evidence of a large source in this region. This is consistent with a recent study based on atmospheric observations and inverse modelling, which found the ESAS region to be a source of only 0.5 to 4.5 Tg y-1 (Berchet et al. 2016).

4.2 Temporal variability of the fluxes

lower than that of GAINS, of 19 Tg y-1.

**30 4.2.1 Seasonal cycle**

Emissions in the HBL region are dominated by wetlands. For this region, our inversion indicates a gradual increase in emissions in spring reaching a maximum in August to September, which is considerably later compared to the LPJ-DGVM model, but close to the LPX-Bern model (Fig. 12). The poorer representation of the seasonal cycle in LPJ-DGVM

Rona Thompson 16/11/2016 16:29 **Deleted:** 8

[revised manuscript text omitted]

Rona Thompson 16/11/2016 16:11

a. Bergamaschi et al. (2013)

**5 b. Bruhwiler et al. (2014)**

Table 4. Summary showing the range of the prior and posterior estimates from scenarios S1 and S2 for the mean fluxes (Tg y-1) for 2005 -2010 (as given in the text) and the trend (Tg y-2) over 2005 - 2013.

|                  | North       | HBL         | Alberta   | North              | WSL                |
|------------------|-------------|-------------|-----------|--------------------|--------------------|
|                  | America     |             |           | Eurasia            |                    |
| prior fluxes     | 9.5 - 14.2  | 2.7 - 4.4   | 1.6 - 3.0 | 43.3 - 44.4        | 11.0 - 12.2        |
| posterior fluxes | 16.6 - 17.9 | 2.7 - 3.4   | 5.0 - 5.8 | 52.5 - 55.5 | 19.3 – 19.9 |
| trend            | 0.38 - 0.57 | 0.22 - 0.23 | 0  | 0.76 - 1.09        | 0           |

Rona Thompson 28/12/2016 15:23 Formatted Table

10

**Figures**

5

**Figure 1**. Map showing the atmospheric measurement sites used in the inversion. The grey dashed line indicates the southern boundary of the inversion domain at 50°N. Flask-air sampling sites are indicated in blue and in situ sites in red.